# RBM20 isoform regulation by independent transcription start sites adapts alternative splicing in development and disease

Michael H. Radke [1,2,11], Victor Badillo Lisakowski[1,2,11], Stefan Meinke [1,2,11], Thiago Britto-Borges[3,4], Valentin Schneider-Lunitz [5], Oliver Hummel[5], Sebastiaan van Heesch [5,6,7], Jorge Ruiz Orera [5], Norbert Hubner [2,5,8,9], Henk Granzier [10], Christoph Dieterich [3,4] & Michael Gotthardt [1,2,9] ✉

RBM20 is a cardiac splicing regulator whose dysfunction causes severe cardiomyopathies. Here, we uncover an unexpected layer of RBM20 regulation through a previously unrecognized transcription start site located between the canonical exon 1 and exon 2. This alternative transcription start site generates a shorter, functional RBM20 isoform translated from an internal ATG in exon 2—identified as the predominant translation start site by ribosome profiling. Despite lacking exon 1, the isoform maintains splicing activity and is conserved across mouse, rat, and human. Strikingly, isoform ratios are tightly controlled during the perinatal period but are selectively altered in disease: in hypertrophic-, unlike in dilated cardiomyopathy, upregulation of RBM20 is driven largely by the alternative isoform. Our findings reveal disease and isoform-specific regulation as a second axis of RBM20 control, operating alongside phosphorylation-dependent nuclear localization, with broad implications for developmental splicing programs, cardiac remodeling, and targeted therapeutic strategies.

Alternative splicing (AS) is a critical mechanism in eukaryotic gene regulation, enabling a single gene to produce multiple mRNA isoforms and thereby diversifying protein function. This process plays a key role in various biological processes, and its deregulation is implicated in cancer, neurodegeneration, and cardiovascular disease. In the heart, AS has been directly linked to functional adaptations during cardiac development, physiological remodeling, and heart disease[1].

The cardiac-enriched RNA-binding protein RBM20 is a key regulator of AS of several essential cardiac genes, including the sarcomeric protein titin and calcium regulators such as *Camk2d* and *Ryr2*[2,3]. Pathogenic variants in the nuclear localization signal within the arginine-serine-rich (RS) domain of RBM20[4], and the glutamate-rich (E-rich) region[5], are associated with aggressive familial forms of dilated cardiomyopathy (DCM) and sudden cardiac death.

[1]Translational Cardiology and Functional Genomics, Max Delbrück Center for Molecular Medicine, Berlin, Germany. [2]German Center for Cardiovascular Research (DZHK) partner site Berlin, Berlin, Germany. [3]Section of Bioinformatics and Systems Cardiology, Klaus Tschira Institute for Integrative Computational Cardiology and Department of Internal Medicine III, University Hospital Heidelberg, Heidelberg, Germany. [4]German Center for Cardiovascular Research (DZHK) partner site Heidelberg/Mannheim, Heidelberg, Germany. [5]Genetics and Genomics of Cardiovascular Diseases, Max Delbrück Center for Molecular Medicine, Berlin, Germany. [6]Princess Máxima Center for Pediatric Oncology, Utrecht, the Netherlands. [7]Oncode Institute, Utrecht, the Netherlands. [8]Helmholtz-Institute for Translational AngioCardioScience (HI-TAC) of the Max Delbrück Center for Molecular Medicine in the Helmholtz Association (MDC) at Heidelberg University, Heidelberg, Germany. [9]Charité Universitätsmedizin, Berlin, Germany. [10]Department of Cellular and Molecular Medicine, University of Arizona, Tucson, AZ, USA. [11]These authors contributed equally: Michael H. Radke, Victor Badillo Lisakowski, Stefan Meinke. ✉e-mail: gotthardt@mdc-berlin.de

As a trans-acting splice factor, RBM20 is involved in the splicing of the giant sarcomeric spring protein titin and leads to the expression of the more adult, stiffer isoform by modulating its I-band region[2]. Knockout models targeting the RNA recognition motif (RRM) of RBM20[6,7] resulted in phenotypes similar to RBM20-deficient rats, where a spontaneous chromosomal deletion removes most of the *RBM20* gene, except for the first exon[2]. Only the rat and the mouse models deficient in exons 4 and 5 of *RBM20*, or with a mutation in the RS domain, display arrhythmia[2,8,9].

In addition to its role in the pathogenesis of heart disease, RBM20 also presents a compelling therapeutic target to modulate cardiac stiffness and calcium handling to improve diastole. Crossing the RBM20 RRM-KO with the titin N2B KO, which does express stiffer titin isoforms, resulted in improved diastolic function in heterozygotes with 50% reduced RBM20 expression[10]. Furthermore, we successfully used *Rbm20* antisense oligonucleotides (ASO) to correct the diastolic dysfunction in the titin N2B KO model[11], and RBM20-ASO dosing was recently optimized to alleviate diastolic dysfunction in a cardiometabolic HFpEF mouse model by selectively increasing compliant titin isoforms[12]. Recent advances in directed therapeutics include base editing to restore splice activity of mutant RBM20[13] and prime editing to correct a DCM-causing RBM20 mutation in human iPSC-derived cardiomyocytes[14].

The splicing activity of RBM20 is regulated via phosphorylation, directing its intracellular and intranuclear localization[15]. Nevertheless, the complexity of *RBM20*'s transcriptional regulation or the existence of functionally distinct isoforms remains poorly understood. Isoform diversity arising from alternative transcription start site (TSS) usage has been shown to regulate RNA processing and to influence cell identity and organism function[16]. Filippello et al. described an independent translation initiation site in Exon 2 of *Rbm20* but were unable to amplify the full-length canonical isoform of RBM20[17]. To investigate whether an independent translation initiation site is utilized independently or if the exon 2 initiated RBM20 isoform shares the canonical transcript start in exon 1, we engineered a *lacZ* reporter gene, including a stop codon, into the end of the translation initiation site in Exon 1 of *Rbm20*. This strategy aimed to disrupt the translation of the canonical RBM20 isoform, effectively generating a knockout model, and to assess whether the alternative isoform, if present, can compensate for the loss. Additionally, the incorporation of the lacZ reporter facilitated the examination of RBM20 expression patterns during development.

Our findings reveal a second independent transcription start site of RBM20, used in mouse, rat and human, which regulates perinatal RBM20 expression. We explored its expression during development and disease, uncovering previously unrecognized transcriptional complexity at the RBM20 locus. These insights suggest that isoform-specific regulation may play a significant role in cardiac development and disease, offering potential avenues for therapeutic intervention.

## Results

### Generation of RBM20 lacZ knock-in reporter mice

We generated a dual-function RBM20 reporter and knockout (KO) mouse strain by replacing the coding sequence after the translation start site in exon 1 with a lacZ-Neo-GK-STOP codon cassette (Fig. 1a, b, Supplementary Fig. 1a). These mice are referred to as RBM20 lacZ. Differentiated ES cell-derived cardiomyocytes contracted and stained blue after the β-Galactosidase (β-Gal) reaction (Supplementary Fig. 1b), confirming lacZ expression under control of the *Rbm20* promoter. RBM20-lacZ embryos at day 13.5 (E13.5) and later stages, as well as adult mice, also stained blue in the heart and skeletal muscle (Fig. 1c, d and Supplementary Fig. 1c, d–g). Western Blot analysis revealed a band at ~180 kDa in heart and skeletal muscle suggesting residual RBM20 expression that involves an alternative isoform, which also varies in expression in different muscles (Fig. 1f, g and Supplementary Fig. 2a).

This shorter RBM20 isoform was detected by immunofluorescence staining in adult heart and skeletal muscle sections of RBM20-lacZ homozygous mice and correctly localizing to the nucleus (Fig. 1h, Supplementary Fig. 2b).

To evaluate RBM20 activity, we analyzed alternative splicing of titin, a primary RBM20 target, via titin gel electrophoresis. Using RBM20 RNA-recognition motif (RRM) deficient heterozygous (RRM-HET) and homozygous (RRM-HOM) KO hearts as size controls[6], we found that the major titin isoform in RBM20-lacZ homozygous hearts is closer to the size of the RRM-HET expressed N-N2BA titin isoform, while RRM-HOM hearts express an even larger giant titin (G-N2BA) isoform (Fig. 1i and Supplementary Fig. 2c, d).

Given the STOP codon at the end of the lacZ gene, fused to the translation start in exon 1, the detection of RBM20 expression and RBM20-mediated splicing activity at RBM20 target genes suggests the presence of additional RBM20 transcription and translation start sites leading to alternative functional isoforms. By 5' RACE PCR and amplicon sequencing (Supplementary Fig. 2e), we identified an additional transcription start site downstream of exon 1, which we designated as exon 1B (Fig. 1j). Because 5' RACE relies on (semi-)nested PCR, band intensities on the agarose gel should be considered semi-quantitative and can over-represent shorter amplicons; accordingly, we use junction counts/PSI values and RT-qPCR for quantitative isoform comparisons (Fig. 1k; Fig. 2e). In addition, the relative abundance of RBM20 transcripts does not necessarily predict proteoform abundance, and our cell-based expression experiments support post-transcriptional regulation of the alternative proteoform output (Supplementary Fig. S2h–j), providing a mechanistic explanation for cases in which exon 1B junction usage appears prominent while the shorter RBM20 band remains weaker on Western blots (Fig. 1f; Supplementary Fig. 2a). Quantitative RT-PCR (qRT-PCR) of exons 1, exon 1B, and the exon 2-3 junction (as an internal control) detected exon 1B expression in wild-type (WT) hearts, with compensatory upregulation in RBM20-lacZ homozygous hearts (Fig. 1k). RBM20 retained nuclear localization and partially colocalized with U2AF65 (Pearson´s R = 0.4) but not SF2/ASF throughout the nucleoplasm and enriched in nuclear speckles across genotypes (Fig. 1l). Beyond the heart, exon 1B was also detected in quadriceps, gastrocnemius and tibialis anterior muscle (Supplementary Fig. 2f). To identify if both isoforms show a similar localization pattern, we co-expressed the EGFP labeled RBM20 canonical isoform and a mCherry labeled RBM20 alternative isoform. Both isoforms display high colocalization (Pearson´s R = 0.6) (Supplementary Fig. S2g). Because the tagged constructs yield robust nuclear signal whereas the untagged alternative construct shows markedly reduced mRNA and protein output (Supplementary Fig. S2h–j), isoform-specific differences in transcript stability, translation efficiency, or proteoform turnover likely shape the effective nuclear dose of RBM20. By overexpression (equal transfection of 1.8 μg plasmid) of RBM20 unlabeled canonical or alternative isoform, we observed that the amount of alternative expressed *Rbm20* is decreased on mRNA level (~10 % of canonical *Rbm20*) and similarly reduced on protein level (Supplementary Fig. S2h-j).

### Increased cardiac compliance in RBM20 lacZ mice

Cardiac performance of homozygous RBM20-lacZ mice (lacZ-HOM) was assessed by echocardiography and catheter-based hemodynamics analysis. Compared to WT controls, lacZ-HOM mice have normal ventricular dimensions but an increased deceleration time and a decreased early diastolic velocity (E') (Table 1), indicating a more compliant left ventricle. These findings align with our previous data on RBM20 downregulation via RBM20 ASO treatment[11]. Additionally, left ventricular end-systolic pressure (LVESP), maximum pressure (Pmax), and pressure development (Pdev) were reduced in homozygous mice (Table 2), as observed for the RBM20 RRM-HOM[7], which also suggests increased ventricular compliance.

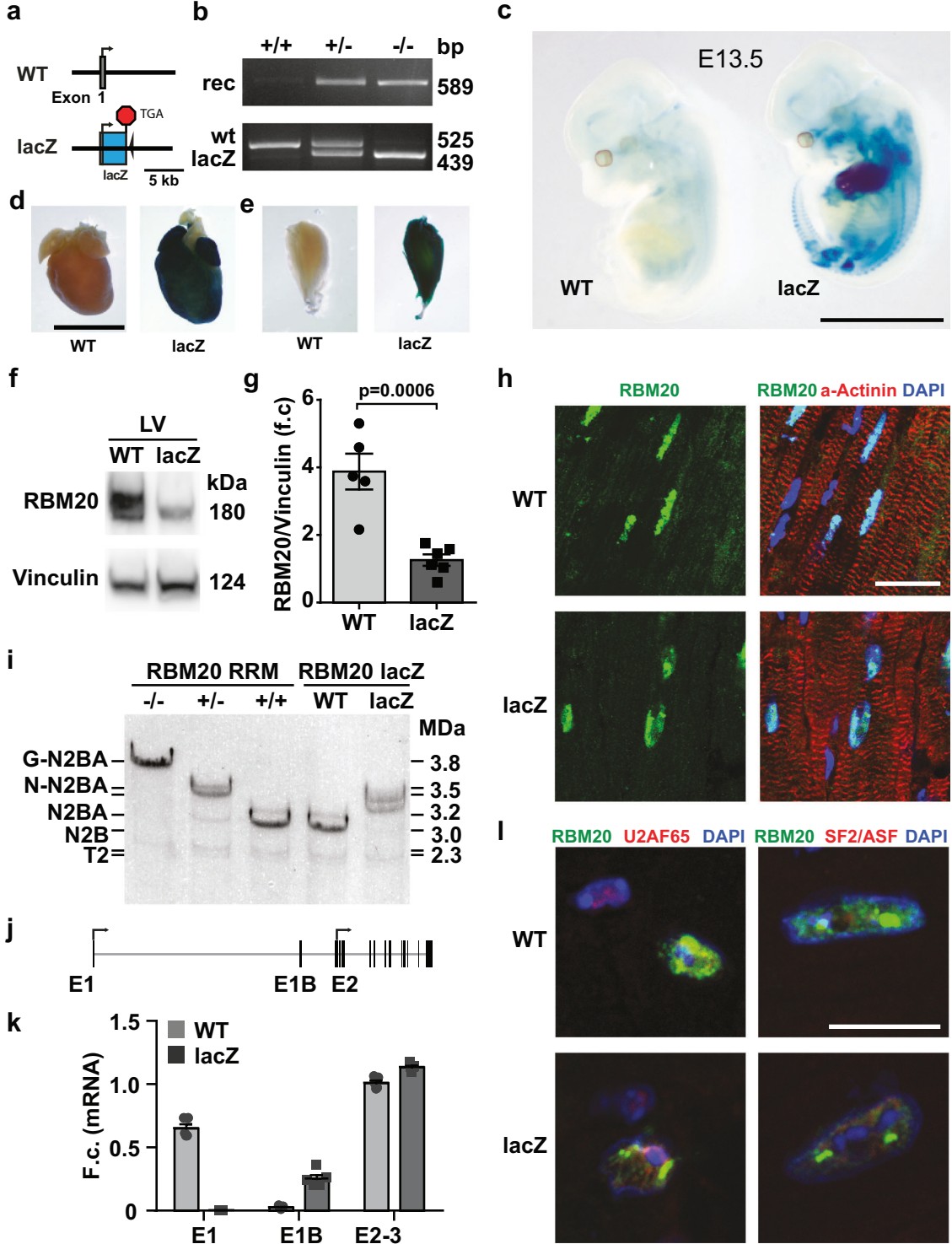

## Characterization of *Rbm20* isoforms in RRM and lacZ mice

To better understand the regulatory mechanisms governing RBM20 isoform expression, we conducted a comparative analysis of RBM20 expression profiles across three genetically engineered mouse models: (1) RBM20-lacZ knock-in mice, (2) heterozygous and homozygous RBM20 RNA Recognition Motif (RRM) knockout mice (with deletions encompassing exons 6 and 7), and (3) wild-type (WT) controls (Fig. 2a). Analysis of previously published mouse heart bulk mRNA-seq and ribosome profiling (Ribo-seq) data[18] confirmed exon 1B expression and identified low levels of ribosome-protected RNA fragments in this region, compatible with ribosome scanning, confirming that the

majority of RBM20 translation initiates at the ATG in exon 2 (Fig. 2b). Ribo-seq data from WT and lacZ-HOM also show only a low P-site number in lacZ-HOM animals in exon 2 upstream of the alternative translation start site, while in WT animals P-site numbers in exon 2 are already high upstream and increase downstream of the alternative translation initiation start (Fig. 2c, d).

Exons 1 and 2 encode compositionally distinct low-complexity segments of RBM20, with a proline-rich (P-rich) region in exon 1 and a leucine-rich (L-rich) region in exon 2. Thus, exon 1B-driven initiation that bypasses exon 1 removes the P-rich N-terminus and shifts the N-terminal proteoform toward an L-rich composition, which could

**Fig. 1 | Generation and validation of Rbm20 exon 1 lacZ mice. a** Replacement of Rbm20 exon 1 with the lacZ gene, incorporating a stop codon (TGA) to disrupt the reading frame. These animals are referred to as RBM20 lacZ. **b** PCR verification of wildtype, heterozygous and knockout/transgene genotype. rec= PCR product after FRT- recombination, wt = wildtype allele PCR product, lacZ= PCR product of the introduced lacZ gene. Genotyping PCRs were repeated >40 independent experiments on over 200 independent samples. **c** lacZ staining of wild-type and knock-out/transgene embryos at E13.5 shows strong staining in the heart and weak staining in the developing skeletal muscles. The experiment was performed 2 times on 2 independent sets of animals. Whole mount lacZ staining of adult (**d**) heart and **e** tibialis anterior (TA) muscle. Staining was performed on 4 independent sets in 3 independent experiments. **f** RBM20 Western blot indicates long and short isoforms in WT hearts, while in the lacZ mice only the shorter one is detected. **g** Quantification of total RBM20 levels from **f**), statistical significance was determined by two-tailed t-test;. Data are presented as mean values +/− SEM. N = 5 WT n = 6 lacZ biological replicates. **h** Immunofluorescence localization of RBM20 in

nuclei of adult WT and lacZ hearts. Staining was performed in 2 independent experiments on 2 independent sets. **i** Titin expression shifts to longer N2BA isoforms in the lacZ mice. RBM20 RRM-KO hearts expressing the giant titin isoform G-N2BA in the homozygote and higher than normal titin isoforms (N-N2BA) in the heterozygous were included for size comparison. Titin gel was performed on 4 independent sets in 2 independent experiments. **j** New exon 1B is mapped based on 5' RACE sequencing. Arrows indicate translation initiation sites. **k** RT-PCR analysis confirms deletion of exon 1 in the lacZ mice and upregulation of the alternative exon 1B. F.c. = fold change. Data are presented as mean value +/− SEM. N = 5 biological replicates per genotype. **l** RBM20 protein is expressed and localizes to the nucleus, with reduced punctate RBM20 protein in the nuclei of RBM20 lacZ cardiomyocytes, where it partially colocalizes with U2AF65 but not with SF2/ASF. Staining was done in 2 independent experiments on 2 independent sets. and Scale bar in **c** and **d** = 5 mm, scale bar in **h** =20 μm, **l** = 10 μm. Source data of **g** and **k** are provided as a Source Data file.

---

tune RBM20 abundance and interactions without splice-regulatory capacity.

The comparison of WT, lacZ and RRM groups revealed the absence of junction reads between exons 1 and 2 and a compensatory upregulation of exon 1B in the lacZ-HOM mice (Fig. 2e). In addition to exons 1 and 1B, we identified an alternative TSS upstream of exon 1, termed exon 1 A, with junction reads connecting to exon 2 (Fig. 2e). We did not detect a distinct exon 1A-derived amplicon band in the 5' RACE gel (Supplementary Fig. 2e), consistent with low abundance and/or limited sensitivity of gel-based RACE for minor TSS events; however, splice-junction reads in RNA-seq provide independent evidence for exon 1A-to-exon 2 connectivity (Fig. 2e). These new TSSs can generate alternative isoforms, designated as alternative A and alternative B (Fig. 2f), with in-frame coding sequences that partially compensate loss of the canonical Rbm20 isoform in the lacZ-HOM mice.

Using exon percent spliced in (PSI) values, we observed similar proportions of exon 1, 1 A, and 1B usage in WT, RRM-HET, and RRM-HOM mice. In contrast, exon 1B was more frequently included than exon 1 in the lacZ-HET, and was almost exclusively used in lacZ-HOM mice (Fig. 2g).

**Transcriptional adaptation in RBM20 lacZ mice at the gene and exon level**

To further characterize transcriptomic differences across genotypes at both gene and exon levels, we performed principal component analysis (PCA) on the gene expression data. Samples clustered according to genotype, with the first principal component (PC1) accounting for 53.8% of the variance (Fig. 3a). Hierarchical clustering based on differentially expressed genes clearly separated RRM-HOM and lacZ-HOM groups from WT and heterozygous genotypes (Fig. 3b). Total Rbm20 mRNA levels were reduced in the RBM20 lacZ-HOM, while remaining unchanged in all other groups (Fig. 3c), indicating that exon 1B upregulation in the alternative B isoform is insufficient to fully compensate for the loss of the canonical isoform at the mRNA level. The number of deregulated genes as compared to WT was 591 in the RRM-HOM mice, 230 in lacZ-HOM and 149 in lacZ-HET, respectively, and only 25 in the RRM-HET mice (Fig. 3d). Notably, 88% of the differentially expressed genes in lacZ-HOM mice were also regulated in the loss-of-function RRM-HOM, suggesting a similar type of adaptation in both models (Fig. 3e and Supplementary Data 1).

At the splicing level, a majority of differentially spliced genes were shared between both homozygous genotypes, with twice as many observed in the RRM-HOM (Fig. 3f, g and Supplementary Data 2). Of the 591 regulated genes in the RRM-HOM, 388 were not shared with lacZ-HOM mice, relating to extracellular matrix organization, tissue maintenance, immune cell migration, and collagen biosynthesis and regulation (Fig. 3h). Among the 91 differentially spliced genes, most were linked to cardiac muscle function, including sarcomere

organization and hypertrophy, ion transport regulation, and mRNA splicing control (Fig. 3i). Of the 45 genes with significant splicing changes in lacZ-HOM mice, 38 overlapped with those in RRM-HOM. Gene ontology analysis identified sarcomere organization as the only enriched term in both gene sets (Fig. 3j, k).

Given this shared enrichment, we examined splicing changes in titin as a central component of the sarcomere and a key RBM20 target. Ttn mRNA displays region-specific sensitivity of alternative splicing depending on the amount of functional RBM20. In the RRM-HOM, splicing was strongly disrupted in the middle and proximal I-band, as well as in the PEVK region of titin (Fig. 3l). RRM-HET mice had similar patterns in the middle I-band and partial inclusions in the proximal PEVK region. In contrast, the lacZ-HOM displayed milder effects, with only the middle I-band affected, while the proximal I-band and PEVK region remained similar to WT levels. LacZ-HET mice also had reduced middle I-band splicing, but closely resembled WT levels overall. These observations suggest a gradient of splice site sensitivity within titin—highest in the PEVK region, followed by the proximal and middle I-band—with the degree of splicing disruption corresponding to the amount of functional RBM20, supporting a dose-dependent regulatory effect. A similar trend is seen for other RBM20 targets, such as Camk2d, Ldb3, and Ttc17 (Supplementary Fig. S3a–c).

**Rbm20 expression is regulated during development**

After confirming splicing activity in the lacZ-HOM mice without canonical RBM20—indicating a functional role for the alternative isoforms—we analyzed isoform usage and regulation across species during development. We utilized a mouse heart developmental multi-omics dataset containing mRNA-seq profiles from embryonic day 10.5 to postnatal weeks 8[19]. When grouping the time points into prenatal, birth, and postnatal stages, total Rbm20 levels increased at birth and then decreased postnatally (Fig. 4a). To examine expression from the three transcription start sites, we quantified splice junction reads from exons 1, 1 A, and 1B to exon 2 and observed a peak around birth for both the canonical exon 1 and alternative exon 1B TSS isoforms, though neither reached statistical significance between the time points (Fig. 4b). Despite similar expression levels, the canonical Rbm20 isoform remained predominant at all time points (Supplementary Fig. 4a). The alternative exon 1 A isoform showed no regulation during development and remained consistently low in abundance, as also observed in WT, RRM, and lacZ mice (Fig. 2). We therefore focused on the canonical exon 1 and alternative exon 1B junction counts for isoform ratio analysis. In mice, this ratio stays constant across the three developmental stages (Fig. 4c). Interestingly, while isoform usage displayed high variability among individual samples before birth and during adulthood, isoform ratios were tightly constrained around the time of birth. This suggests that the fetal-to-adult transition period is a critical window in which RBM20 isoform balance is under stringent

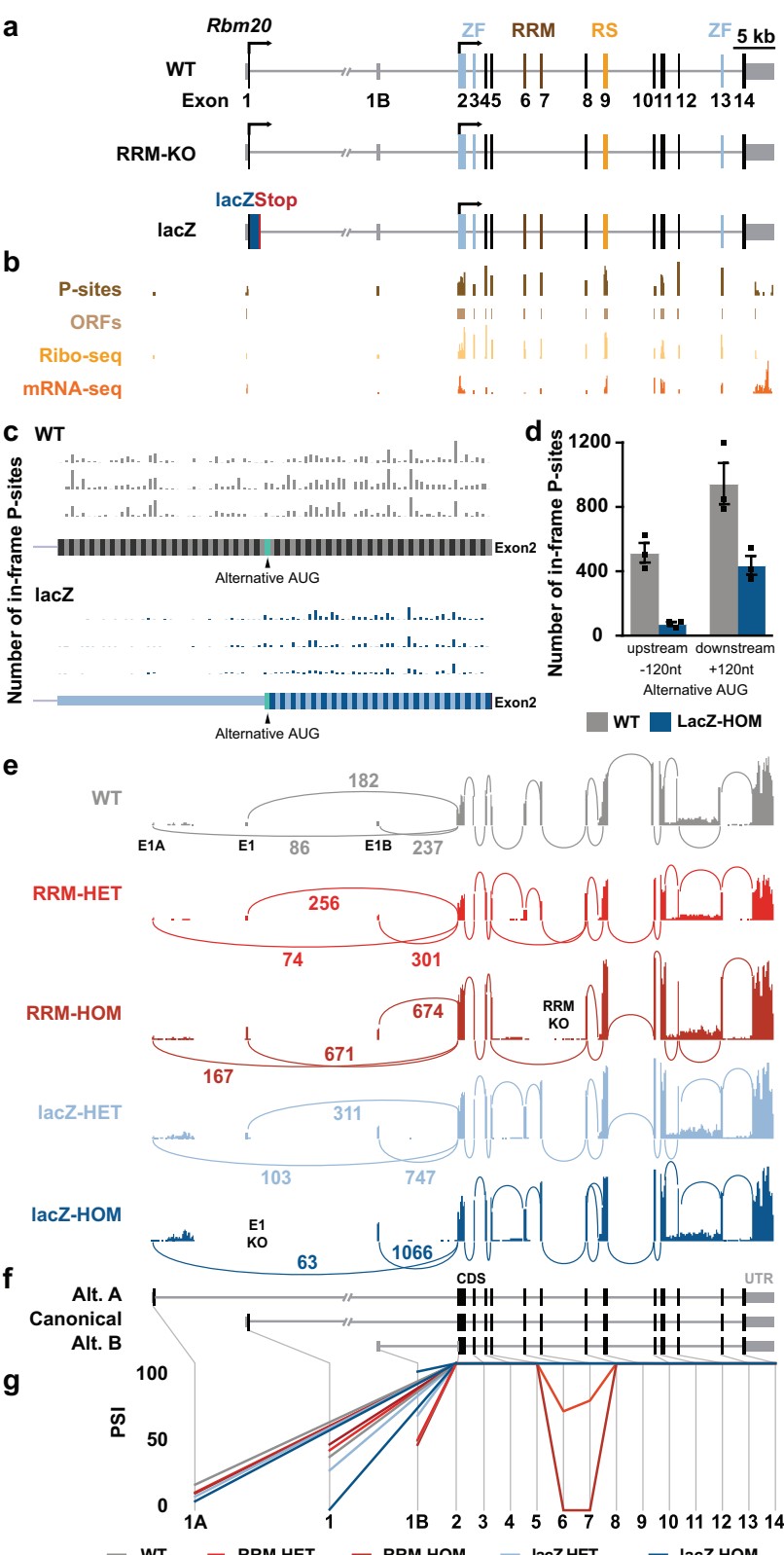

**Fig. 2 | Identification of alternative RBM20 isoforms. a** *Rbm20* exon structure of WT, RBM20 RRM-KO and RBM20 lacZ knock-in mice. Arrows indicate translation initiation sites. **b** Alternative *Rbm20* transcription starting site (TSS) in intron 1 is detected by mRNA-seq and Ribo-seq. **c** P-sites of Ribo-Seq analysis of exon 2 from WT and RBM20 lacZ-HOM. **d** Number of upstream and downstream P-sites in WT and LacZ-Hom exon 2. $N = 3$ for WT and $n = 3$ for lacZ-HOM mice in c and d, data are presented as mean values +/− SEM (**e**) Sashimi plot reveals 2 alternative *Rbm20* transcription start sites (exon 1 A and exon 1B). **f** Transcript structure of the canonical and alternative (Alt. A and Alt. B) isoforms. **g** Percent spliced-in (PSI) plot of *Rbm20* in RRM-HET, RRM-HOM, lacZ-HET and lacZ-HOM mice. Exons are marked by boxes and introns by grey lines. $N = 3$ for WT and RRM-HET, $N = 4$ for RRM-HOM, lacZ-HET and lacZ-HOM in panels **c**, **e**. ZF: zinc finger motif, RRM: RNA recognition motif, RS: arginine/serine-rich domain, nt: nucleotides. Source data of d) is provided as a Source Data file.

**Table 1 | Echocardiography of WT – RBM20 lacZ**

| | WT | RBM20 lacZ |
|---|---|---|
| N | 7 | 8 |
| age (w) | 14 ± 0 | 14 ± 0 |
| body weight (g) | 23.5 ± 0.6 | 23.4 ± 0.5 |
| LV mass (mg) | 116.3 ± 8.4 | 127.1 ± 7 |
| heart/body weight | 5.0 ± 0.4 | 5.4 ± 0.2 |
| Echocardiography | | |
| heart rate (bpm) | 377 ± 13 | 333 ± 16 |
| stroke volume (µl) | 23.2 ± 2 | 21.4 ± 2 |
| Vol dia (µl) | 53.8 ± 4.1 | 57.8 ± 3.4 |
| Vol sys (µl) | 30.6 ± 3 | 36.1 ± 2 |
| LV dia (mm) | 4.2 ± 0.18 | 4.4 ± 0.12 |
| LV sys (mm) | 3.33 ± 0.2 | 3.58 ± 0.14 |
| LVPW dia (mm) | 0.75 ± 0.03 | 0.74 ± 0.04 |
| LVPW sys (mm) | 0.95 ± 0.06 | 0.88 ± 0.05 |
| IVS dia (mm) | 0.74 ± 0.03 | 0.76 ± 0.04 |
| IVS sys (mm) | 0.96 ± 0.05 | 0.98 ± 0.07 |
| FS (%) | 21.0 ± 1.8 | 18.8 ± 1.9 |
| EF (%) | 43.6 ± 2.6 | 36.8 ± 1.9 |
| Mitral Doppler | | |
| E (mm/s) | 610 ± 48 | 541 ± 68 |
| A (mm/s) | 366 ± 22 | 331 ± 32 |
| E/A ratio | 1.67 ± 0.1 | 1.63 ± 0.11 |
| IVRT (ms) | 22.6 ± 1.8 | 31.8 ± 3 p = 0.025 |
| IVCT (ms) | 20.9 ± 1.8 | 34.7 ± 2.6 p = 0.001 |
| MVDT (ms) | 21.8 ± 1.9 | 29.6 ± 2.5 p = 0.029 |
| ET (ms) | 50.3 ± 1.5 | 52.3 ± 1.6 |
| E' (mm/s) | 21.1 ± 1.6 | 15.2 ± 2 p = 0.044 |
| A' (mm/s) | 16 ± 1.6 | 14.1 ± 1.5 |
| E' / A' | 1.3 ± 0.1 | 1.1 ± 0.1 |
| E / E' | 30.3 ± 3.5 | 38.3 ± 4.5 |
| Cardiac Performance | | |
| CO (ml/min) | 8.8 ± 1 | 7.1 ± 0.7 |
| MPI | 0.86 ± 0.05 | 1.27 ± 0.06 p < 0.001 |

Statistical significance determined in a two-tailed Student´s *t*-test.
*LV* left ventricle, *Vol.* volume, *dia.* diastole, *sys.* systole, *LVPW* left ventricle posterior wall, *IVS* interventricular septum, *FS* fractional shortening, *EF* ejection fraction, *IVRT* isovolumetric relaxation time, *IVCT* isovolumetric contraction time, *MVDT* mitral valve deceleration time, *ET* ejection time.

**Table 2 | Pressure volume relationship of WT – RBM20 lacZ**

| | WT | RBM20 lacZ |
|---|---|---|
| N | 7 | 8 |
| age (w) | 14 ± 0 | 14 ± 0 |
| body weight (g) | 23.5 ± 0.6 | 23.4 ± 0.5 |
| HR [BPM] | 463 ± 17 | 444 ± 13 |
| Pressure/ Volume | | |
| MAP [mmHg] | 89.2 ± 1.9 | 78 ± 4 p = 0.031 |
| LVESP [mmHg] | 104.4 ± 2.6 | 94.7 ± 3.3 p = 0.041 |
| LVEDP [mmHg] | 9.0 ± 2 | 7.2 ± 1.1 |
| Pmax [mmHg] | 108.2 ± 2.4 | 100.1 ± 2.4 p = 0.034 |
| Pmin [mmHg] | 4.9 ± 1.2 | 4.6 ± 1.2 |
| Pmean [mmHg] | 45.6 ± 1.7 | 40.4 ± 1.4 p = 0.030 |
| Pdev [mmHg] | 103.4 ± 3 | 95.5 ± 2.8 |
| LVESV [µL] | 15.6 ± 1.9 | 17.2 ± 3.2 |
| LVEDV [µL] | 34 ± 1.7 | 33.2 ± 3.3 |
| SV [µL] | 19.6 ± 0.9 | 18.5 ± 2 |
| CO [ml/min] | 9.0 ± 0.5 | 8.2 ± 0.9 |
| CI [ml/min/BW] | 386.8 ± 26.3 | 350.1 ± 39.3 |
| Ea [mmHg/µL] | 5.4 ± 0.2 | 5.6 ± 0.7 |
| TPR | 10.1 ± 0.6 | 10.3 ± 1 |
| EF [%] | 58.1 ± 3.2 | 55.2 ± 3.2 |
| dP/dt max [mmHg/s] | 8034 ± 527 | 7516 ± 408 |
| SW [mmHg*µl] | 1902 ± 146 | 1665 ± 171 |
| -dP/dt min [mmHg/s] | 9679 ± 875 | 8234 ± 393 |
| TauWeiss [ms] | 7.8 ± 0.7 | 7.8 ± 0.4 |
| ESPVR [mmHg/µl] linear | 4.8 ± 0.9 | 3.8 ± 1 |
| EDPVR [mmHg/µl] linear | 0.2 ± 0 | 0.2 ± 0 |
| PRSW [mmHg] | 52.4 ± 6 | 41.6 ± 3.9 |

Statistical significance determined in a two-tailed Student´s t-test.
*HR* heart rate, *MAP* mean arterial pressure, *LVESP* left ventricle end-systolic pressure, *LVEDP* left ventricle end-diastolic pressure, *P* pressure, *LVESV* left ventricle end-systolic volume, *LVEDV* left ventricle end-diastolic volume, *SV* stroke volume, *CO* cardiac output, *CI* cardiac index, *Ea* arterial elastance, *TPR* total peripheral resistance, *EF* ejection fraction, *SW* stroke work, *ESPVR* end-systolic pressure-volume relationship, *EDPVR* end-diastolic pressure volume relationship, *PRSW* preload recruitable stroke work.

control. Disruption of this control could perturb developmental splicing programs and contribute to disease susceptibility later in life.

To determine if the expression of more than one *Rbm20* isoform is unique to the mouse or conserved across species, we analyzed RNA-seq data from rat hearts. Here, two 5′ TSSs directly connect with exon 2, which are homologous to the canonical and alternative B isoforms in mice (Fig. 4d). Exon 1B reads were detected in both mRNA-seq and Ribo-seq data from rat left ventricle tissue in an independent dataset[18] (Supplementary Fig. 4b). We therefore refer to the rat *Rbm20* isoforms as canonical and alternative. In rat heart mRNA-seq data from embryonic day 19 to postnatal day 10, total *Rbm20* levels peak around birth (P1), which is accompanied by an increased trend in alternative isoform levels, while the canonical exon 1 remains stable during these time points (Fig. 4e, f). The isoform ratio is only regulated at P10 (Fig. 4g), mostly due to a decrease of exon 1B usage.

**Regulation of RBM20 isoforms in cardiac disease**
Given the association of *RBM20* pathogenic variants with certain types of cardiomyopathies such as dilated (DCM)[20] and hypertrophic

cardiomyopathy (HCM)[21,22], we used mRNA-seq data from spontaneously hypertensive rats (SHR) in comparison to Brown Norway (BN) controls to analyze RBM20 expression in a hypertrophy model. Total *Rbm20* mRNA levels were increased in SHR (Fig. 5a), with a similar elevated trend in both isoforms (Fig. 5b). Similar to the developing mouse and rat hearts, the canonical isoform remains predominant in both adult SHR and BN controls. However, the canonical/alternative ratio is decreased in SHR, which suggests differential regulation of the two RBM20 transcription start sites, potentially reflecting independent promoter activity (Fig. 5c).

To explore whether elevated RBM20 expression in SHR hearts leads to changes in splicing of known targets, we compared all differentially spliced genes in SHR heart tissue to a curated list of 45 established RBM20 targets (Supplementary Data 6)[3,23,24]. Surprisingly, only 7 of these targets (including CamK2D but not titin) were differentially spliced in SHR (Fig. 5d and Supplementary Data 3), suggesting that despite increased RBM20 levels, its splicing activity may be functionally saturated or constrained in the adult heart. This finding supports the idea that RBM20-mediated splicing is not solely dosage-dependent but may also require isoform-specific regulation or co-factors for full activity.

To explore the expression of alternative human *RBM20* isoforms and their relevance to heart failure, we first analyzed mRNA-seq[25] and ChIP-seq[26] data from adult control human hearts. An alternative TSS

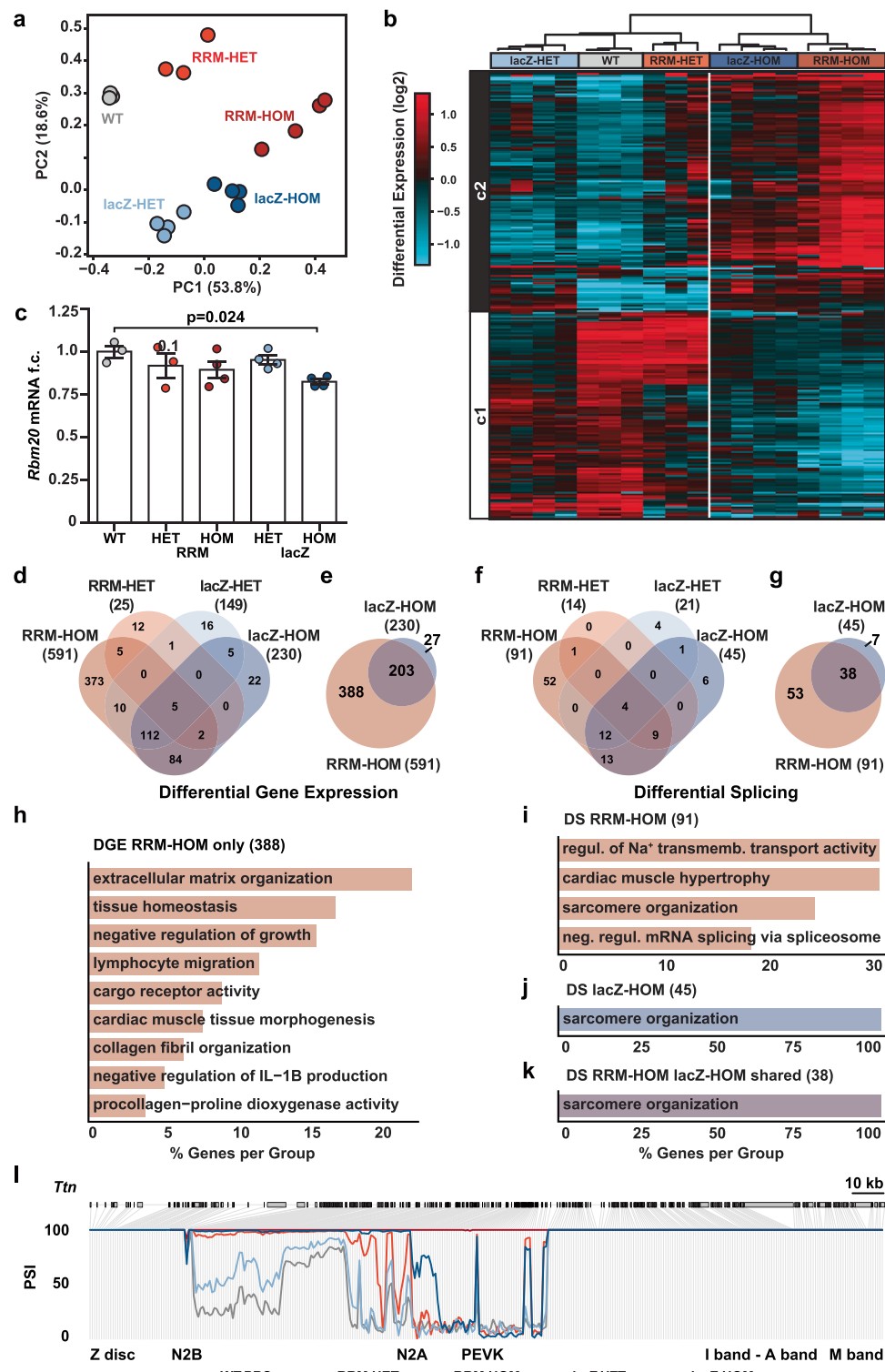

**Fig. 3 | Differential gene expression and splicing analysis in RRM-KO and lacZ knockin animals. a** Principal component analysis (PCA) of all analysed genotypes in mRNA-seq data. **b** Heatmap and hierarchical clustering by differential gene expression between WT, heterozygous and homozygous RRM-KO (RRM-HET and RRM-HOM) and heterozygous and homozygous lacZ (lacZ-HET and lacZ-HOM). **c** *Rbm20* mRNA expression level normalized to wildtype. Data are presented as mean values +/- SD. Statistical significance was determined with one way ANOVA p = 0.03 with Tukeys post test. **d** Venn diagram of shared and exclusive differentially expressed genes between all analysed genotypes and **e** only between RBM20 RRM and lacZ-HOM (**f**) Venn diagram of differential spliced genes of all analysed genes. **g** Venn diagram of differential spliced genes between RRM-HOM and lacZ-HOM. **h** Gene Ontology analysis of differentially expressed genes only in the RRM-HOM knockout. **i** Gene ontology by Biological Process of differentially spliced genes of RRM-HOM, **j** lacZ-HOM or **k** shared between RRM-HOM and lacZ- HOM. **l** PSI plot of titin exons in all analysed genotypes. Transcript structure is represented above the trace. Exons are marked by boxes and introns by grey lines. N = 3 for WT and RRM-HET, N = 4 for RRM-HOM, lacZ-HET and lacZ-HOM. Source data of **c** is provided as a Source Data file.

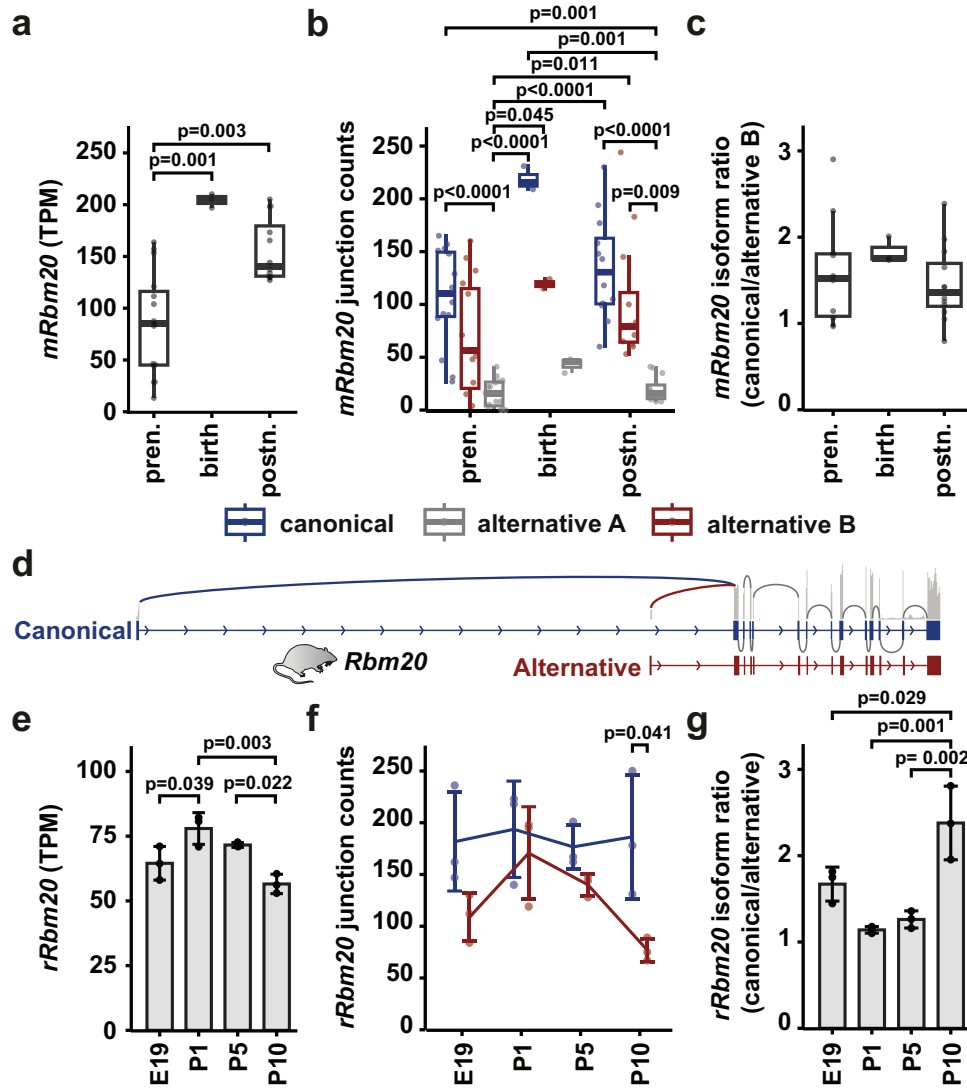

**Fig. 4 | Rbm20 isoform usage is regulated during development. a** *Rbm20* expression level in prenatal (pren, $n = 15$), newborn (birth, $n = 3$) and postnatal (postn, $n = 12$) development of mouse. **b** *Rbm20* junction reads spanning from exon 1 to exon 2 in prenatal vs. birth vs. postnatal samples based on reads mapping to the canonical (blue), alternative B (red), or alternative A (grey) transcription start. Prenatal $n = 15$, birth $n = 3$, postnatal $n = 12$. **c** Isoform ratio derived from **b**. Prenatal $n = 11$, birth $n = 3$, postnatal $n = 12$. **d** *Rbm20* gene structure in rat. **e** *Rbm20* expression level around birth in the rat. **f** *Rbm20* expression levels around birth based on reads mapping to the canonical (blue), alternative B (red) in rat. **g** Isoform ratio derived from **f**. Data in **a**–**c** are presented as median with interquartile range; whiskers extend to the most extreme data points within 1.5× the interquartile range. Outliers were identified per developmental stage for each subfigure using the 1.5×IQR rule and removed prior to analysis. Data in e-g are presented as average with standard deviation. TPM: transcripts per million. Sample sizes are: for mouse 15 prenatal (E10.5 – E18.5 combined), 3 birth (P1), and 12 postnatal (PW1 – PW8 combined) samples, for rat 3 samples per developmental stage. Statistical significance was determined using the Kruskal-Wallis test followed by Dunn's post hoc test for multiple comparisons for **a**–**b**, statistical significance was determined using ANOVA (two-sided) with Tukey´s post-test for **c**, **e**–**g**. Source data are provided as a Source Data file.

homologous to rat and mouse exon 1B that directly splices to exon 2 is also expressed in humans, at comparable levels to the canonical isoform (Fig. 5e). Additionally, H3K4me3 ChIP-seq signal around both TSSs indicates active transcription. We also detected human *RBM20* exon 1B reads in mRNA-seq and Ribo-seq data from an independent study[18] (Supplementary Fig. 5a).

We then analyzed mRNA-seq data from an HCM cohort with 97 patients and 23 controls[27]. Total *RBM20* is also increased in HCM (Fig. 5f); intriguingly, this increase is driven almost entirely by upregulation of the alternative isoform, while canonical isoform levels remain unchanged (Fig. 5g). This results in a decreased isoform ratio (Fig. 5h), similar to that observed in SHR with cardiac hypertrophy, indicating that the appropriate maintenance of RBM20 isoform production is critical for cardiac homeostasis, and that this regulatory mechanism is transcriptionally controlled. In

HCM patients, 17 out of 45 potential RBM20 targets were differentially spliced, out of a total of 612 differentially spliced genes (Fig. 5i and Supplementary Data 4).

Overall, more RBM20 targets are spliced in human HCM (38%) compared to the SHR rat (16%) (Fig. 5d, i Supplementary Fig 5d and Supplementary Data 4;). The splicing of Titin was not affected (Supplementary Fig. 5b). Gene Ontology enrichment of the differentially spliced genes in HCM highlighted cytoskeletal and adhesion modules, including cell-adhesion molecule binding, actin binding, focal adhesion, and cell-cell junction terms (Supplementary Fig. S5d). These enrichments align with the concept that hypertrophic remodeling emphasizes mechanotransduction and junctional adaptation, and they support the interpretation that RBM20-linked splice remodeling in HCM concentrates on actin-adhesion-junction networks rather than on titin isoform switching.

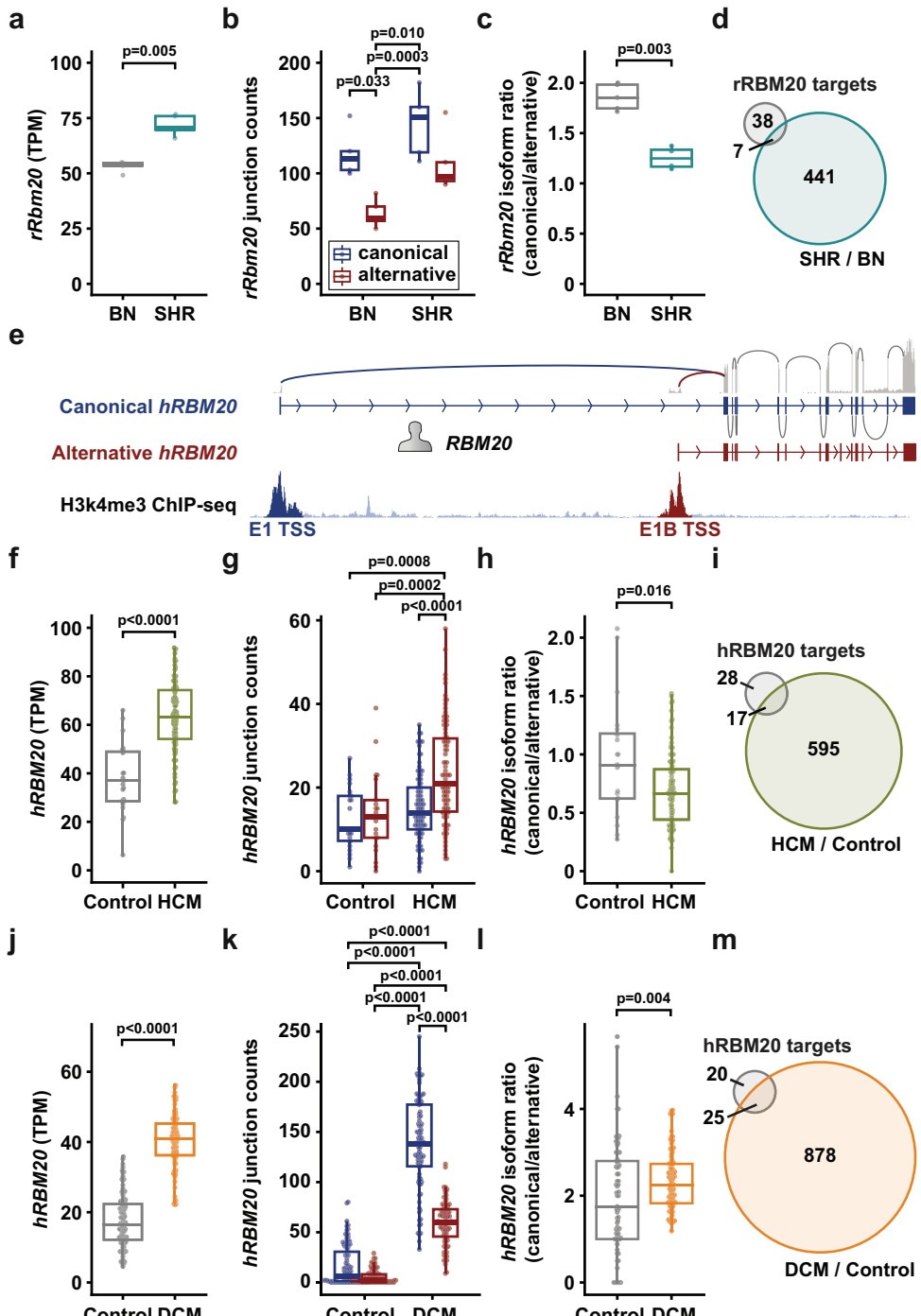

**Fig. 5 | RBM20 isoform regulation in cardiomyopathy. a** Total *Rbm20* mRNA expression in Brown Norway (BN) wildtype and spontaneously hypertensive rats (SHR). **b** *Rbm20* junction reads spanning from exon 1 to exon 2 in BN and SHR samples based on reads mapping to the canonical (blue), alternative B (red) transcription start. **c** Rat *Rbm20* isoform ratio derived from **b**). **d** Venn diagram of differentially spliced genes and overlap to RBM20 spliced genes. **e** *RBM20* gene structure in human, below: ChIP-seq reads mapping to human *RBM20* exon1 and 1B. **f** Total *RBM20* mRNA levels in non-failing (control, *n* = 23) and HCM (*n* = 97) hearts. **g** Junction counts to exon 2 for canonical exon1 and alternative exon1B (Control *n* = 22, HCM *n* = 96). **h** Isoform ratio of *RBM20* in control (*n* = 20) and HCM (*n* = 96) conditions. **i** Venn diagram of differentially spliced genes in HCM/control and overlap to RBM20 spliced genes. **j** Total *RBM20* mRNA levels in non-failing (control, *n* = 105) and DCM (*n* = 92) hearts. **k** Junction counts to exon 2 for canonical exon1

and alternative exon1B (Control *n* = 103, DCM *n* = 96). **l** Isoform ratio of *RBM20* in control (*n* = 73) and DCM (*n* = 95) conditions. **m** Venn diagram of differentially spliced genes in DCM/control and overlap to RBM20 spliced genes. Outliers were identified per status group for each subfigure using the 1.5×IQR rule and removed prior to analysis. Data are presented as median with interquartile range; whiskers extend to the most extreme data points within 1.5× the interquartile range. Statistical significance was determined using the Kruskal-Wallis test followed by Dunn's post hoc test for multiple comparisons for **a–c**, **g–h**, **j–l**, or two-sided ANOVA with Tukey´s post test for **f**. Significance threshold for differentially spliced genes in **d**, **i**, **m** FDR < 0.05 and difference in mean PSI of 10%. DCM: dilated cardiomyopathy, HCM: hypertrophic cardiomyopathy, TPM: transcripts per million. Sample size for rat BN-SHR are 5 samples each. Source data are provided as a Source Data file.

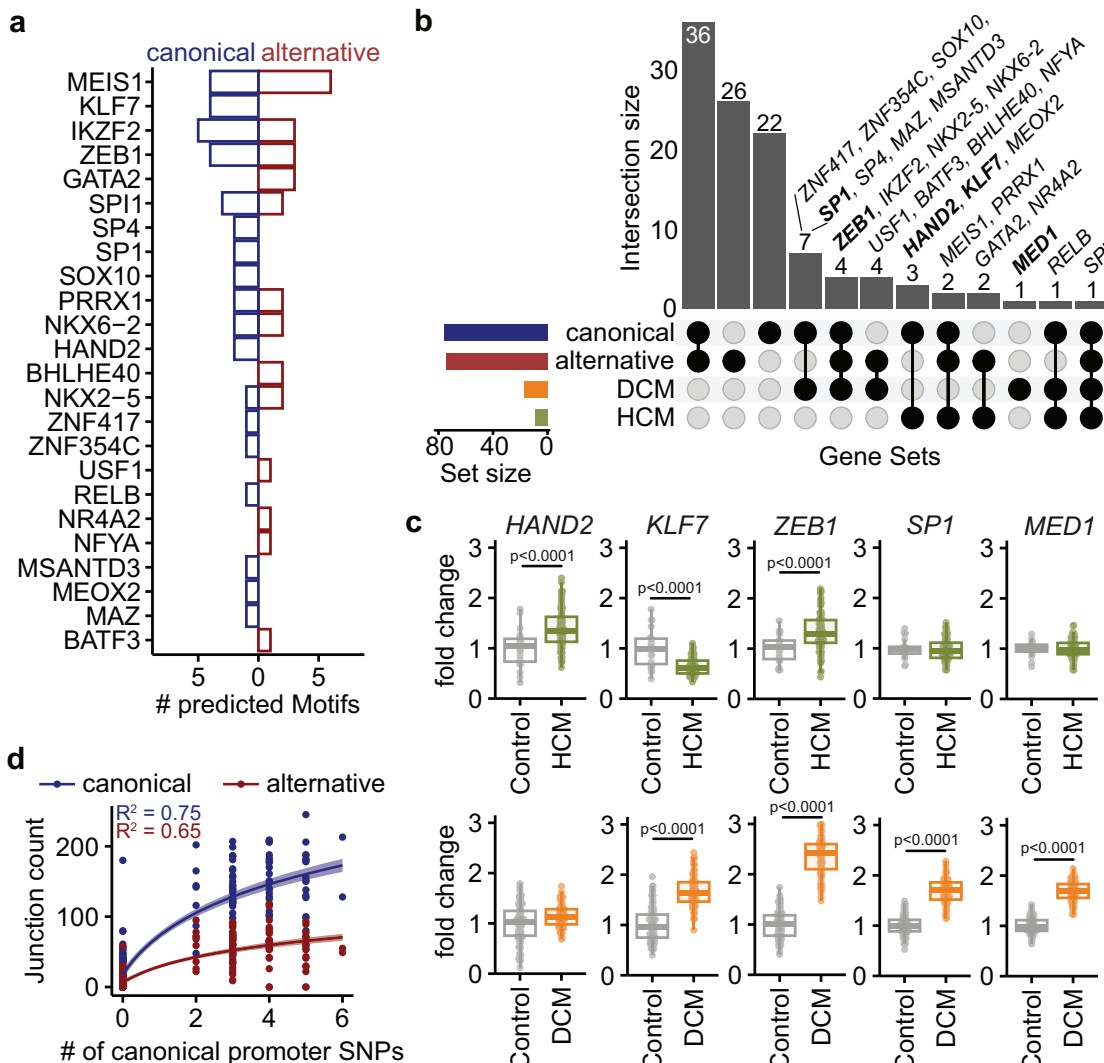

**Fig. 6 | Transcriptional regulation of RBM20 isoform expression in human cardiomyopathy. a** Number of predicted TF binding motifs within the canonical exon 1 or alternative exon 1b promoter sequence, which includes 2000 bp upstream and 100 bp downstream of each exon start. TF motif scan was performed using the JASPAR database (https://jaspar.elixir.no/) selecting all TFs and filtered for high-confidence binding motifs (relative score = 1), in addition, TF genes were filtered to be differentially expressed comparing either DCM vs. Control or HCM vs. Control or which have ChIP peaks in at least one promoter region (based on chip-atlas.org, 29.07.2025, *MED1, HAND1, CTCF, ESRRG, TBX5*). **b** Upset plot showing the overlap of high-confidence TFs and TFs that are differentially expressed comparing either DCM vs. Control (DCM) or HCM vs. Control (HCM) (DESeq2-derived padj <0.01 and absolute log2FC > log2(1.5)). **c** Differential TF expression in DCM and HCM. Boxplots show the distribution of fold-change values per group. The center line indicates the median; boxes represent the interquartile range (25th–75th

percentiles); whiskers extend to the most extreme data points within 1.5× the interquartile range. Outliers were identified per Gene × status group using the 1.5×IQR rule and removed prior to analysis. DESeq2-derived adjusted p-values are shown (Wald-tests; Benjamini-Hochberg correction, two-sided), Control (HCM) $n = 20$-23, HCM $n = 91$-96, Control (DCM) = 103-106, DCM $n = 92$-95. Gene x status specific sample numbers are summarized in the Source Data file. **d** Regression analysis of canonical promoter SNP burden in DCM samples and RBM20 junction counts. Lines represent linear regression fits of junction counts as a function of log-transformed canonical promoter SNP counts (log(x + 1)). Shaded areas indicate 95% confidence intervals of the fitted regression. Adjusted $R^2$ values from the linear models are shown. SNP burden in canonical promoter is associated with increased RBM20 expression in DCM samples, especially the canonical isoform. R2: adjusted coefficient of determination DCM: dilated cardiomyopathy, HCM: hypertrophic cardiomyopathy. Source data are provided as a Source Data file.

In DCM, upregulation of *RBM20* in comparison to non-failing controls has been previously reported (Fig. 5j)[25]. Using the same dataset, we examined isoform usage and found that both isoforms are upregulated in DCM patients, with the canonical isoform showing the strongest increase (Fig. 5k). This is reflected in a modestly elevated isoform ratio (Fig. 5l). Analysis of differentially spliced genes in the DCM cohort revealed that 25 out of 45 (56%) known RBM20 targets were differentially spliced (Fig. 5m, Supplementary Fig 5e and Supplementary Data 5). Gene Ontology enrichment in DCM emphasized muscle differentiation and contractile architecture (myofibril, contractile fiber, muscle tissue development) together with RNA splicing and transcription coregulator activity (Supplementary Fig. S5e),

consistent with a broader remodeling program that couples sarcomere re-patterning to global transcript-processing adaptation. Despite elevated *RBM20* levels in HCM and especially in DCM, *TTN* splicing was not majorly affected (Supplementary Fig. 5c). This is expected, since the shorter, fully spliced N2B isoform is the predominant one in the adult left ventricle[28], leaving few or no additional splice sites for RBM20 to act on.

## Disease-specific transcription factor programs and promoter variants shape RBM20 isoform expression

To investigate regulatory mechanisms underlying differential RBM20 isoform usage in cardiomyopathy, we analyzed transcription factor

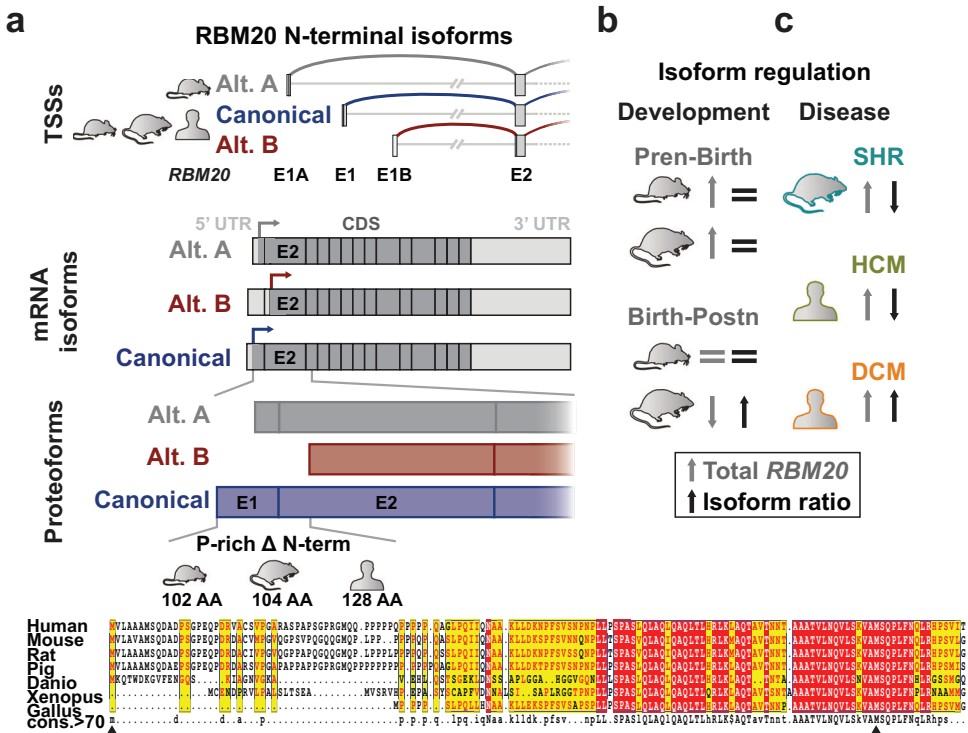

**Fig. 7 | Landscape of RBM20 isoform regulation. a** Identified *RBM20* transcription start sites (TSSs), and the corresponding canonical and alternative mRNA and predicted protein isoforms (proteoforms). In comparison to the shorter, alternative B proteoform, the canonical one retains a P-rich N-terminal amino acid stretch in mammals but the region around alternative translation start is highly conserved between species. Arrows indicate translation initiation sites. **b** *Rbm20* expression and isoform ratio in mouse and rat development (Pren – prenatal, Postn – postnatal) and **c** in the spontaneously hypertensive rat (SHR) and in patients with hypertrophic cardiomyopathy (HCM) or dilated cardiomyopathy (DCM). AA: amino acid.

(TF) motifs and promoter variation at the canonical and alternative RBM20 TSS. TF binding motif analysis of promoter regions (2 kb upstream and 100 bp downstream of the exon start site) identified distinct sets of high-confidence TF binding sites (Fig. 6a). Several of these TFs, including HAND2, KLF7, ZEB1, SP1, and MED1, were either differentially expressed in DCM or HCM or showed ChIP-seq evidence of promoter binding (Fig. 6b). Expression analysis confirmed disease-specific regulation of these TFs, with HAND2 up- and KLF7 down-regulated in HCM, whereas ZEB1, SP1, and MED1 were significantly induced in DCM (Fig. 6c). These factors are known regulators of cardiac development, growth, and remodeling: HAND2 is essential for heart morphogenesis[29], MED1 and SP1 are global transcriptional regulators implicated in cardiomyopathy[30,31], ZEB1 has been linked to cardiac remodeling and fibrosis[32], and KLF7 regulates cardiomyocyte metabolism and hypertrophy[33]. These results suggest that distinct TF programs may preferentially activate the alternative promoter in HCM versus the canonical promoter in DCM.

To further assess *cis*-regulatory influences, we performed RNA-seq-based variant calling in promoter regions. SNP burden within the canonical promoter was positively correlated with RBM20 junction usage in DCM samples, particularly for the canonical isoform (Fig. 6d). Together, these data indicate that alternative isoform expression in HCM is likely driven by a disease-specific TF landscape favoring the alternative promoter, while canonical isoform upregulation in DCM reflects both TF activity and *cis*-regulatory variation at the canonical promoter.

## Discussion

Our study uncovers a previously unrecognized layer of regulatory complexity at the RBM20 locus by demonstrating that multiple transcription start sites (TSSs) generate RBM20 mRNA isoforms with distinct 5′ ends and protein N-termini. These isoforms are dynamically regulated during cardiac development and are remodeled in cardiomyopathy, suggesting an intricate regulatory mechanism to fine-tune splicing control in the heart[34].

The RBM20-lacZ reporter allele, which disrupts the exon-1-initiated transcript, unexpectedly retained detectable RBM20 protein expression and splicing activity. Using 5′ RACE and RNA-seq, we identified two alternative first exons within intron 1 (exon 1 A and exon 1B), consistent with independent promoters. Importantly, ribosome profiling supports that the dominant RBM20 translation initiation occurs at the AUG in exon 2, such that transcripts initiating at exon 1B encode a shorter, N-terminally truncated RBM20 proteoform. In lacZ homozygous hearts, exon 1B usage increases and partially compensates for loss of the exon 1 transcript; correspondingly, splicing of key RBM20 targets such as Ttn and Camk2d remains largely preserved but is less efficient.

In cell-based assays, fluorescently tagged canonical and alternative RBM20 showed comparable nuclear localization and high spatial overlap when co-expressed. Notably, the alternative construct accumulated at lower steady-state levels after transient transfection, suggesting additional regulation at the mRNA and/or protein level. Together with the lower abundance of the alternative RBM20 band on Western blots despite similar (or higher) transcript abundance in sequencing datasets, these observations point to post-transcriptional control of proteoform output[35].

RBM20 contains N-terminal low-complexity segments enriched for proline (exon 1) and leucine (exon 2), but the field currently lacks a definitive mechanistic assignment for these regions. Recent syntheses of RBM20 domain architecture emphasize that these compositional elements exist alongside the canonical RNA-binding and RS-rich modules that dominate RBM20 localization and activity[20]. In parallel,

multiple lines of evidence link splicing regulators with low-complexity regions to biomolecular condensates and nuclear speckle biology[36], and RBM20 itself forms discrete nuclear splicing foci in cardiomyocytes. In this context, our imaging data suggest that both canonical and alternative RBM20 localize to nuclear puncta and co-localize when co-expressed (Supplementary Fig. S2g), consistent with RS-domain governed nuclear trafficking and foci association as dominant determinants of subnuclear targeting[37]. By contrast, our transient overexpression experiments show that the alternative construct produces substantially lower mRNA and protein levels (approximately 10% of canonical, Supplementary Fig. S2h–j).

Cross-species analysis indicates that exon 1B-driven isoforms are present in rat and human, and that the region surrounding the exon 2 AUG is conserved across vertebrates (Fig. 7a). Initiation at exon 2 yields a shorter proteoform lacking an N-terminal stretch of approximately 50–128 amino acids, depending on the species. The functional consequences of this truncation remain to be determined and could involve differences in protein–protein interactions, stability, folding, or post-translational modification.

Both canonical and alternative mRNA isoforms peak around birth in mouse and rat (Fig. 7b), coinciding with the perinatal switch of titin isoforms from fetal N2BA to adult N2B[38,39]. We speculate that increased RBM20 transcription at this time window supports efficient remodeling of RBM20-dependent splicing programs. Finally, the variable RBM20 isoform distribution we observe across skeletal muscles suggests that promoter usage and/or RBM20 dosage may contribute to tissue-specific titin splicing outcomes[40].

Our comparative analysis between the RBM20-lacZ and RBM20-RRM-KO mice revealed that the presence of alternative isoforms in the lacZ mice partially compensates for the loss of the canonical isoform, as evidenced by intermediate titin splicing patterns and cardiac compliance measures. In contrast, the RRM-KO model, lacking all functional RBM20 isoforms, had more pronounced splicing defects and cardiac dysfunction[7]. Together, these findings support a model in which alternative promoter usage provides a second axis to tune effective RBM20 activity in vivo. This is in line with the findings that many human genes are not regulated only by a single promoter[41]. Furthermore, alternative splicing can be influenced by the promoter[42].

Current reference annotations still underrepresent RBM20 5′ complexity: for example, RefSeq curates a single reviewed RBM20 mRNA/protein record while listing additional computationally predicted transcript variants. Our identification of conserved alternative 5′ initiation that yields a functional proteoform therefore closes an annotation gap and argues that isoform-specific promoter usage constitutes a regulated layer of RBM20 biology rather than transcriptional noise. Although many minor TSS events can reflect stochastic initiation, the evolutionary conservation and regulated developmental and disease-associated isoform ratio shifts support functionality in this locus[43].

In cardiac pathology, such as HCM and DCM, we observed differential regulation of *RBM20* isoforms. In HCM, the alternative isoform is predominantly upregulated, whereas in DCM, both canonical and alternative isoforms show increased expression. Interestingly, despite elevated *RBM20* levels (Fig. 7c), titin splicing changes were minimal, consistent with a saturation effect in the adult heart, where all modifiable splice sites are already fully utilized in the adult heart.

HCM-associated splicing changes enrich for actin and adhesion/junction ontologies (Supplementary Fig. S5d,e), consistent with remodeling of mechanotransduction and structural coupling. In contrast, DCM enriches for muscle differentiation and contractile architecture together with RNA splicing and transcription co-regulator activity, suggesting broader coupling between sarcomere re-patterning and transcript-processing adaptation in ventricular failure.

Canonical and alternative RBM20 form overlapping puncta in the nucleus, indicating that the alternative N-terminus preserves nuclear targeting and the propensity to partition into RBM20-positive nuclear compartments (Supplementary Fig. S2g). This observation argues against a localization switch as the main functional difference and instead supports isoform-specific regulation of effective nuclear dose, interaction topology, or condensate dynamics[36,37].

Altogether, our findings highlight the importance of transcriptional and translational diversity in RBM20 regulation. This motivates a testable therapeutic hypothesis: strategies that tune RBM20 should consider both total nuclear RBM20 dose and isoform composition, because dose-dependent splice-site sensitivity can create region-specific outcomes in RBM20 targets[3]. Future in vivo studies should determine whether selectively shifting promoter usage or proteoform balance offers a safer therapeutic window than uniform RBM20 suppression. These insights have therapeutic implications: modulating RBM20 levels or isoform usage should consider the dynamic balance between isoform expression, splicing efficiency, and saturation thresholds in adult myocardium.

Future research should focus on elucidating the cis-regulatory elements governing alternative TSS usage in *RBM20*, the functional differences between its isoforms, and the interplay between RBM20's expression, post-translational modifications, and splicing activity. Such insights will be crucial for developing precise interventions aimed at correcting splicing defects in cardiomyopathies associated with RBM20 dysfunction[10,11,44].

## Methods

All experiments involving animals were performed according to institutional guidelines and had been approved by the local authorities (LAGeSo Berlin).

### Generation of RBM20 knock-out with lacZ knock-in mice

The generation of the RBM20 knockout with fusion of the lacZ gene to the translation start site of RBM20 was done via a targeting vector (Fig. 1) using standard procedures[45]. The animals were backcrossed on a 129/S6 background after successful integration.

### Genotyping

Genomic DNA was prepared from mouse ear biopsies with the HotSHOT method[46]. The genotypes of RBM20 (Primer: fwd, GAGAAGGAC AAGGGGACTGG, WT rev CAAAAATTATGCCCCACCAC, KO/KI rev CCGTAATGGGATAGGTCACG) were determined by PCR and visualized on agarose gels.

### Animal procedures

Mice were kept at the animal facility of the MDC in individually ventilated cages and a 12 h day and night cycle with free access to food and water. We used age-matched mice at 10–15 weeks and included only male animals. We randomized animals to experimental groups where applicable and blinded investigators during acquisition and analysis for echocardiography and catheter measurements as described earlier[10].

Sex-dependent differences in cardiac maturation and remodeling may influence RBM20 promoter usage and proteoform output. Although we used age- and sex-matched adult mice for physiological phenotyping, we did not power this study for genotype-by-sex interaction testing; therefore, we did not perform formal sex-stratified analyses. Future studies should prospectively test whether sex modifies RBM20 isoform ratios and RBM20-dependent splicing outcomes during development and in disease contexts.

### Histology of heart and skeletal muscle

Beta Galactosidase staining: Mice were sacrificed and perfused with 4% paraformaldehyde (PFA) in PBS. After overnight fixation, the tissue was washed in rinse buffer (100 mM sodium phosphate; 2 mM MgCl2; 0.1% TritonX-100) for 30 min at room temperature. Staining was performed

overnight at 37 °C in staining solution (5 mM potassium ferricyanide; 5 mM potassium ferrocyanide; 20 mM Tris pH 7.3; 1 mg/ml XGal in rinse buffer). After a post fixation in 4% PFA at 4 °C the tissue was washed 3 times for 15 min in PBS. Embryos were cleared as described earlier[47]. Pictures were taken on a Zeiss SteREO Discovery V8 microscope. Embryos were cleared as described earlier[47].

## Transfection of cells

For colocalization experiments HEK293 (ACC-305; Lot 28) cells were grown on 0.1% gelatin coated glass coverslips and transfected either with EGFP-RBM20 canonical, mCherry-RBM20 alternative or both plasmids using Pei160. Plasmids were generated on the pEGFP-C1 (clontech) backbone, by inserting a cDNA-generated hRBM20 PCR product. In the mCherry-RBM20 plasmid, EGFP was replaced by mCherry, from pmCherry-C1 (TaKaRa). All Plasmids were sequenced prior to transfection. 72 h after transfection cells were fixed and imaged with an 63X oil immersion objective on a Leica SP8 confocal microscope. Images were processed with Leica LAS Software. For expression of RBM20 isoforms, we transfected 1.8 µg of myc- tagged human RBM20 canonical or RBM20 alternative plasmids to HEK-EBNA cells with Pei160. 48 h after transfection, cells were harvested from a 6 well plate and mRNA and protein were prepared and used for RT-PCR and Western blot detection of RBM20 expression.

## Molecular analysis

**RT-PCR**. Real-time SYBR green PCR was performed with PowerUp SYBR (Thermo Fischer Scientific) according to the manufacturer׀'s instructions. RNA for Real Time TaqMan analysis was prepared with RNeasy Plus Micro Kit (Qiagen), cDNA was generated with the high-capacity RNA-to-cDNA Kit (Applied biosystems) and TaqMan run was performed with TaqMan Gene Expression Master Mix (Applied biosystems) according to the manufacturer´s instruction. For normalization we used 18S. Information on primers and amplicons is listed in Supplementary Table 1. PCR was performed on a QuantStudio6 Pro (Applied biosystems).

**Titin gel**. To separate titin isoforms we used vertical SDS agarose gel electrophoresis (VAGE) as previously described[48].

**Western blot**. Proteins were separated on an SDS-PAGE. Western blot was performed on PVDF membranes. The anti-RBM20 (self-made)[2], Vinculin/Tubulin and c-Myc antibodies are listed in Supplementary Table 2. The secondary HRP-conjugated antibody was detected by chemiluminescence staining with ECL (Supersignal West Femto Chemiluminescent Substrate; Pierce Chemical Co.) on a FusionFX system, and quantification of blots was performed with AIDA software v 4.19.

**IF- staining and microscopy**. Frozen heart and tibialis anterior tissue were sectioned at 7 µm thickness with CryoStar NX70 cryostat, fixed in 4% PFA, washed twice in PBS and blocked with goat serum. The sections were incubated over night at 4 °C with anti-RBM20 (self-made)[2] and a-actinin primary antibodies, washed five times in PBS and incubated for 2 h at room temperature in the dark with 1:1000 4′,6-diamidino-2-phenylindole (DAPI) staining and secondary antibodies. The slides were mounted with Dako Fluorescence Mounting Medium (Agilent) and imaged with 63X oil immersion objective on a Leica SP5 confocal microscope. Images were processed with Leica LAS Software. Antibodies are listed in Supplementary Table 3. Colocalization was determined with Fiji coloc2 software.

## Transcriptomic analysis

mRNA-seq data analysis from WT, RRM-KO and lacZ mouse left ventricles was conducted with Illumina TrueSeq® Stranded mRNA Library prep and indexes kits conducted to manufacturer's instructions and as previously described[11]. Hierarchical clustering by differential gene expression using cosine similarity and PCA plots were generated with AltAnalyze v2.1.4.3[49]. Leafcutter v0.2.9[50] was chosen for differential splicing analysis. Differentially spliced events were filtered by adjusted p value < 0.01 and a difference in percent spliced-in (dPSI) of ±0.2. PSI plots were generated as described previously[51]. Gene Ontology analysis was performed with Cytoscape v.3.9.0[52] and the ClueGO v2.5.9[53] app. Venn diagrams were generated with Venn Diagram Plotter v.1.6 or https://bioinformatics.psb.ugent.be/webtools/Venn/ when comparing more than 3 groups. Sashimi plots were extracted with Integrative Genomics Viewer 2.16[54].

The additional left ventricle mRNA-seq datasets analysed comprised the following studies: for dilated cardiomyopathy (DCM), data from Heinig et al. (2017)[25] were used (European Genome-Phenome Archive: EGAD00001003390 and EGAD00001003391); for hypertrophic cardiomyopathy (HCM), datasets from Garmany et al. (2024) were employed[27]; mouse developmental data were obtained from Gu et al. (2022)[19]. For the rat data, RNA was isolated using TRIzol Reagent (Invitrogen; 15596018) using 10 mg tissue. Poly(A)-purified mRNA-seq libraries were generated according to the TruSeq mRNA Reference Guide, using 500 ng of total RNA as input. Libraries were multiplexed and sequenced on an Illumina HiSeq 2000 producing paired 2x101nt reads. Mouse RRM-KO[6] and lacZ samples were sequenced with 2x150nt reads.

**RNA-seq processing and quantification.** Adapters and low-quality reads were removed from the raw sequences using fastp (v0.23.2)[55] with the options -D and -c, enabling for deduplication and base correction in overlapped paired-end regions. Quality-filtered reads were aligned to species-specific reference genomes using STAR (v2.7.8a)[56] with primarily default settings, but specifying options for output formatting and quantification, including −outSAMtype BAM SortedByCoordinate, --quantMode GeneCounts, and --seedPerWindowNmax 15. Alignments were performed against custom annotations for human (GRCh38.111), mouse (GRCm39.111), and rat (RatBN7.2.111), which included all Ensembl-annotated transcripts and the manually curated alternative *RBM20* transcripts, specifically incorporating the alternative A and alternative B start exons (derived from mRNA and Ribo-seq data[18]). For quantification of *RBM20* transcript expression, we extracted reads mapping across the junction between the respective first exon and exon 2 from the STAR-generated splice junction files (*SJ.out.tab). Exact genomic coordinates are provided in Supplementary Table 4.

Transcript-level quantification was performed using Salmon (v1.10.1)[57] in the quasi-mapping-based mode. The index was built from the curated transcriptomes, and quantification was run with the following options: --validateMappings for improved mapping accuracy, -l A to enable automatic library detection. Gene-level expression estimates were generated in R using the tximport v1.30.0[58] package, applying the scaledTPM method to aggregate transcript-level abundances to the gene-level while accounting for transcript length and sequencing depth.

**Ribo-seq data processing and mapping.** Ribosome profiling (Ribo-seq) reads were processed following standard preprocessing steps[59], using three biological replicates each WT and RBM20 lacZ-HOM samples. First, we trimmed Ribo-seq datasets to remove residual adapter sequences using TrimGalore v0.6.6. Ribo-seq datasets were further filtered to exclude common contaminants, including mitochondrial, rRNA, and tRNA sequences. We mapped all Ribo-seq files to the mouse Ensembl genome and transcriptome (release 98; GRCm38/mm38) using STAR v2.7.3a with the following parameters: --outSAMtype BAM SortedByCoordinate, --outFilterMismatchNmax 4, --outFilterMultimapNmax 20, --alignSJDBoverhangMin 6, --alignSJoverhangMin 500, --outFilterType BySJout, --limitOutSJcollapsed 10000000, --limitIObufferSize 300000000, and

--outFilterIntronMotifs RemoveNoncanonical. Calculated P-site positions from Ribo-seQC v0.99.0 were intersected with annotated Rbm20 coding sequences, and differences in P-site density between WT and lacZ-HOM samples were quantified within the Rbm20 locus, including ±120 nucleotides surrounding the alternative AUG codon in exon 2.

**Differential expression.** Differential gene expression was assessed with DESeq2 (v1.42.1)[60] using STAR-derived gene counts. To reduce noise from low-abundance transcripts, genes were retained for analysis only if they had at least 10 raw counts in at least half of the samples prior to DESeq2 normalization and differential expression testing. Log2 fold changes were adjusted using the apeglm (v1.24.0) shrinkage method[61] to stabilize effect size estimates. Genes were called differentially expressed at padj <0.01 and |log2FC| > log2(1.5).

**Alternative splicing.** Exon-level percent spliced-in (PSI) values were calculated using the PSI Python scripts from https://github.com/MIAOKUI/PSI[62]: exon inclusion counts were generated with dexseq_count.py (options -p yes -s reverse -r pos -f bam), exon exclusion counts with exclusion_count.py, and PSI values with psi_calculation.py (specifying -l 100) using Python 3.10.7. Alternative splicing events were additionally identified using rMATS-turbo (v4.3.0)[63] using the options -t paired --readLength 100 --variable-read-length --libType fr-firststrand. Splicing events were called significant at FDR < 0.05 and |ΔPSI| > 10%. Gene Ontology enrichment was performed with the packages clusterProfiler (v4.10.1)[64] and org.Hs.eg.db (v3.20.0) setting the pvaluecutoff = 0.05 and ont = "ALL".

**Transcription factor motif analysis.** For transcription factor (TF) motif binding analysis, the JASPAR database (https://jaspar.elixir.no/) was used to scan the human canonical and alternative *RBM20* promoter sequences ( + 2000 bp upstream and + 100 bp downstream of the exon start site) for TF motifs (Homo sapiens, performed on 25 July 2025). Only high-confidence motifs (relative score = 1) were retained. In addition, ChIP-seq data from the ChIP-Atlas database (chip-atlas.org, selecting H. sapiens (hg38), Track type class = ChIP: TFs and others, Cell type Class = Pluripotent stem cell, Threshold for Significance = 50, performed on 29 July 2025) were used to identify TFs with promoter occupancy found in iPSC-derived cardiac cells. TFs predicted by JASPAR and/or present in ChIP-Atlas were considered in downstream analyses.

**Variant calling.** Variant calling was performed on RNA-seq alignments using freebayes (v1.3.9, https://github.com/freebayes/freebayes)[65] with the indexed human reference genome (GRCh38, chromosome 10). Analyses were restricted to the *RBM20* canonical promoter region (chr10:110642335–110644435) and alternative promoter region (chr10:110762614–110764714). Resulting variant call format (VCF) files were compressed and indexed with samtools/htslib (v1.19). Sample identifiers in VCF headers were corrected with bcftools reheader (v1.14), and VCFs were merged per group (bcftools merge). Promoter SNP burden was subsequently correlated with *RBM20* isoform expression in R. All subsequent data analyses were performed in R (v4.3). Ensembl ID to gene name mapping was performed using the biomaRt package (v2.61.1), VCF files were imported and parsed using the Bioconductor package VariantAnnotation (v1.52.0), and subsequent data processing and visualization were performed using the tidyverse (v2.0.0) suite of packages.

**Statistics**
Statistical analysis was done with the GraphPad Prism Software (Version 5) or R (v4.3), including the package dunn.test (v1.3.6). Outliers were identified using the interquartile range (IQR) method, defining values as outliers if they fell below the first quartile minus 1.5 times the

IQR or above the third quartile plus 1.5 times the IQR. Statistics for heart development, mouse[19], HCM[27], and DCM datasets[25] were calculated in R using the Kruskal-Wallis with Dunn's post-hoc test for not normally distributed samples based on Shapiro test. For normally distributed samples statistical significance was assessed using a two-sided t-test or ANOVA with Tukey´s post-test.

**Reporting summary**
Further information on research design is available in the Nature Portfolio Reporting Summary linked to this article.

## Data availability
The mouse RRM-KO, lacZ, and rat data generated in this study have been deposited in the European Nucleotide Archive (ENA) at EMBL-EBI under accession code PRJEB89457. In addition, we analyzed the following publicly available datasets: DCM and Control samples available in European Genome-Phenome Archive under accession code EGAD00001003390 and EGAD00001003391, HCM and Control samples available on the NIH Gene Expression Omnibus (GEO) under the accession number GSE249925, and the mouse developmental data deposited in the GEO repository under the accession number GSE213233. Source data are provided with this paper.

## Code availability
Computer code to generate the results is available on GitHub (https://github.com/MitchGotthardt/GotthardtLab/tree/main).

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

## Acknowledgements

We thank the MDC platforms for their support with generating and analyzing the animal model: the transgenic facility for generation of the knock-in model, the genomics facility for sequencing support, the phenotyping facility for echocardiography and catheter analysis, the Microscope Core Facility for support with the confocal microscopes. We are grateful to Carmen Judis, Janine Fröhlich and Susanne Blachut for expert technical assistance. We thank Vita Dauksaite for generation of the EGFP-RBM20 canonical Plasmid.

## Author contributions

M.G., M.R., V.B.L., N.H., and S.v.H. planned experiments; M.R. and V.B.L. performed experiments. M.R., V.B.L., and S.M. analyzed data with support from T.B.B., V.S.L., J.R.O., S.v.H., O.H. and C.D.; N.H., S.v.H., and V.S.L. generated the RNA-seq data from rat and human samples. HG provided the RBM20 RRM knockout mouse model and contributed to data interpretation. M.G., M.R., V.B.L., and S.M. wrote the manuscript with input from all authors.

## Funding

This work was funded by the European Research Council (ERCAdv to MG), the Leducq Foundation (CASTT to MG) and the German Research Foundation (DFG; SFB-1470 Project B04 and Z01 to MG), the German Research Foundation (DFG; SFB-1470 Project B03 to NH), the British Heart Foundation and Deutsches Zentrum für Herz-Kreislauf-Forschung (BHF/DZHK; SP/19/1/34461 to NH), Pathfinder Cardiogenomics Programme of the European Innovation Council of the European Union (DCM-NEXT; to NH) and by the DZHK (German Centre for Cardiovascular Research; to NH and MG). Open Access funding enabled and organized by Projekt DEAL.

## Competing interests

The authors declare no competing interests.
