## [Transparent Peer Review file · Nature Communications]

RBM20 isoform regulation by independent transcription start sites adapts alternative splicing in development and disease

Corresponding Author: Professor Michael Gotthardt

Version 0:

Reviewer comments:

Reviewer #1

(Remarks to the Author)

The authors of this manuscript identified a alternative transcription start site in canonical exon of RBM20, a well known splicing regulator. Mutation in RBM20 cause cardiomyopathies and understanding the regulation and function of Rbm20 of relevance The manuscript is well written, the data sound and novel. Addressing a few comments would further increase the quality of the manuscript.

1) Validation of translation initiation at the ATG at exon 1B. The data presented in figure 2b are not completely convincing. The authors could include additional ribosomal sequencing experiments using harringtonine to better map the initiation on the ATGs and include ribosomal sequencing data from the lacZ-Hom mice to show translation initiation.

2) What is the function of the P-rich n-terminal region in the canonical isoform. The authors speculate the it may be more efficient at organizing RBM20 into nuclear speckles. Functional data supporting this hypothesis would strengthen the manuscript.

3) What drives upregulation of the alternative isoform in HCM vs upregulation of the canonical form in DCM? Could the authors use existing data on HF patient to look at regulatory elements that might explain alternative TSS usage in different forms of HF?

Reviewer #2

(Remarks to the Author)

In this manuscript, Radke et al. report the identification of a novel transcription start site in the RBM20 gene, leading to the production of a shorter protein isoform. Using RNA-seq technology and reanalysis of previously published datasets, the authors demonstrate that this isoform is expressed in both rat and human tissues, retains splicing activity, and exhibits differential regulation compared to the canonical isoform in the context of human disease. The manuscript is generally well written; the Methods section is thorough and appropriately detailed, and the figures are clear and support the authors' conclusions. While the discovery of this novel isoform provides valuable insight into an additional layer of RBM20 regulation, the study lacks experimental evidence directly addressing the biological significance or functional impact of the new isoform.

Minor Comments

1. There is a repeated paragraph between lines 71–73 and lines 78–81.
2. Figure 1b is not referenced in the main text.
3. The terminology used to refer to the different genetic models is somewhat inconsistent and occasionally confusing. It should be standardized throughout the manuscript to improve readability and interpretation

Major Comments

1. In Figure 1f and Supplementary Figure 1a, the WT samples show two bands, with the smaller band appearing to match the size of the one detected in the lacZ mouse. Do the authors interpret this band as representing the alternative Rbm20 isoform? If so, what is the rationale for emphasizing the functional significance of this isoform if it is also present in WT animals?
2. In Figures 1h and 1l, it is unclear whether the nuclear localization patterns of the canonical and alternative isoforms are similar or distinct. The authors should clarify this point.

3. In Figure 1i, the authors suggest that the two bands observed in the lacZ lane correspond to those in the RRM+/- mouse. However, the bands in the lacZ sample appear clearly lower in molecular weight. This discrepancy should be addressed.
4. In Supplementary figures 1c and 1d, the authors make similar claims regarding band identity and size, but appropriate control samples (e.g., RRM+/-) are missing for comparison.
5. Table 1 indicates improved ventricular output in the LacZ-HOM mouse. However, it is unclear whether this phenotype is due to reduced expression of the canonical RBM20 isoform or the gain of function from the alternative isoform. The authors should discuss this distinction more explicitly.
6. In all presented Ribo-seq datasets, the translational signal for the alternative Rbm20 isoform is higher than that of the canonical isoform. How do the authors reconcile this with their observation that the canonical isoform is more abundant than the alternative in most models except the LacZ-HOM?
7. If the values shown above the arcs in the Sashimi plot represent the number of splicing events observed, how do the authors explain that the value supporting the alternative isoform in WT samples is higher than that for the canonical isoform? This seems to contradict other expression data presented.
8. Although the authors present a PSI plot, it remains unclear whether they are claiming that the alternative RBM20 isoform regulates a distinct Ttn splicing pattern compared to the canonical isoform. This should be clarified with supporting data or discussion.
9. Performing Gene Ontology analysis on the differentially spliced genes shown in Figures 5i and 5l would provide further insight into the molecular pathways through which the alternative RBM20 isoform may contribute to cardiomyopathy.
10. While RBM20 is predominantly expressed in cardiomyocytes, it would be valuable to validate the findings shown in Figure 5 using publicly available human single-cell RNA-seq datasets for hypertrophic and dilated cardiomyopathy. This could help exclude the possibility that the observed effects are driven by non-cardiomyocyte populations.
11. The manuscript presents convincing data regarding the expression of the alternative Rbm20 isoform. However, its functional role in cardiac development and disease progression remains insufficiently explored. This limits the overall impact of the study. Additional functional experiments or a more thorough discussion of its biological relevance would be beneficial.

Reviewer #3

(Remarks to the Author)

The manuscript by Racke et al. with the title „RBM20 isoform regulation by independent transcription start sites adapts alternative splicing in development and disease“ identified an unrecognized transcription start site (TSS) for RBM20 via genetic perturbations in mice. The study convincingly shows that the ratios of the different RBM20 isoforms are controlled during development and are contrarily altered in DCM versus HCM. The authors furthermore focus on newly and already published sequencing data from mice, rat and humans, and provide novel insights into the landscape of RBM20 isoform regulation.

The manuscript contains the information that is required to understand the conducted experiment, the used approach is well described and allows others to perform additional mechanistic studies also in other disease-relevant contexts. To fully agree to a publication in Nature Communications, I suggest the following optimizations:

Major:

1. The authors used a self-produced RBM20 antibody, which is not described nor referenced throughout the manuscript. Therefore, not enough detail is provided for the work to be reproduced. Ideal would have been a comparison to already commercially available RBM20 antibodies.
2. The introduction as well as the abstract state strongly that RBM20 is regulated by phosphorylation. However, it is then neglected through the manuscript. The authors see a size shift of the shorter RBM20 1B isoform, but don't state if the observed size shift is expected by an exclusion of the first exon. The deletion should be around 128 amino acids (~13 kDa), however it seems that the shift on the provided Western blot is bigger than that. A quick search at the phosphosite plus database indicates two potential phosphoserines in the first part of RBM20 for humans, could that explain the bigger than 13 kDa shift? The authors also neglect that the first exon includes the L-rich domain of RBM20, the manuscript only states that the P-rich domain is deleted in 1B.
3. The authors show convincingly that the regulation of isoforms is not mediated via translational regulation, and assume that it comes from a transcriptional regulation via alternative transcription start sites. The authors then suggest quickly that the ratio maybe regulated by two independent promoters for both transcript start sites. This concept needs some further details to be fully incorporated into the main text. However, could it also be that the integration of the LacZ gene into Exon1 disrupts normal splicing of RBM20 itself and therefore a new splice isoform with only Exon 2 is incorporated via a non-canonical new splice site? This should be quickly discussed as an alternative possibility or experimentally excluded in the main text.
4. In general, the manuscript does not describe prominent observed bands on provided Western blots and agarose gels, which should be discussed or contextualized. The authors state that Filippello et al. already described an independent translation initiation site in Exon 2 of Rbm20, but also state that Filippello did not detect the here called full-length canonical isoform of RBM20 in human cardiac tissues. The here provided RACE PCR results in supplementary Fig. 2e show that the shorter fragment seems to be the predominant form as well, which disagrees to the other shown data in the main figures. This needs further clarification. In addition to exons 1 and 1B, the authors also identified an alternative TSS upstream of

exon 1, termed exon 1A, however the shown RACE PCR gel results do not detect a longer variant, which should be discussed in the main text too. In general, all shown western blots and agarose gels should indicate the ladder/size at the side! The Western blot in Fig. 1f shows only the bigger band (please provide the full picture, and show also the shorter bands), the sashimi plot in Fig. 2c indicates the shorter version should be predominant. The Western blot in supplementary Fig. 2a also shows mostly that the bigger band is predominant. This needs some better explanations in the main text

5. Supplementary Fig. 2a shows different RBM20 bands in the provided Western blot, which indicates different isoforms distributions in the isolated muscle tissues, which needs to be further discussed in the main text. The LacZ mice seem to differ more between tibialis anterior muscle and quadriceps than the shown WT mice.

6. Fig.1 I needs quantification of the colocalization to make the claim that there is colocalization of RBM20 and U2AF65!

7. The introduction mentions that the new identified isoform of RBM20 might offer potential avenues for therapeutic intervention, without further details on possible intervention points in the main text.

Minor:

1. No reference in the main text to Fig 1b
2. All figure legends should contain and explain all used abbreviation, e.g. what is "rec" in Fig. 1b
3. Fig. 1K shows that exon 2-3 is as highly expressed in LacZ as in WT! Can the authors comment on the here observed compensation mechanism?
4. All figure legends should indicate the number and age of the used mice or the corresponding used data sets, e.g. Fig. 2b does not indicate this, but it is required to understand at which time point the alternative Rbm20 transcription starting site (TSS) in intron 1 is detected by mRNA-seq and Ribo-seq. Same for Fig. 1h, the Immunofluorescence localization of RBM20 in nuclei of WT and lacZ hearts does not indicate the age.
5. What is the difference between "RiboSeq" and "Psite" data in Fig. 2?
6. In Supplementary Fig. 2f, what does the axis label "Norm to H" mean?
7. The material and methods part is quite sparse in regard of the performed mRNA seq procedure and could benefit from a more detailed description of used materials/kits.
8. Figure legend 4 e should be labeled with "f" and not "e"
9. The method section states that the study was performed only on male mice, it would have been interesting to see if there is a sex difference in the RBM20 isoform ratio. Why has this not been considered?

Reviewer #4

(Remarks to the Author)

Version 1:

Reviewer comments:

Reviewer #1

(Remarks to the Author)

The authors addressed all my concerns and the overall quality of the manuscript improved.

Reviewer #2

(Remarks to the Author)

The authors have made a commendable effort to address all the comments raised by this reviewer. With the inclusion of the newly generated data, and the corresponding adjustments to the text, the manuscript has been substantially improved, and I believe the revised version is now suitable for publication in Nature Communications.

Reviewer #3

(Remarks to the Author)

The authors have addressed all comments and concerns and I recommend the manuscript for publication.

Reviewer #4

(Remarks to the Author)

I co-reviewed this manuscript with one of the reviewers who provided the listed reports. This is part of the Nature Communications initiative to facilitate training in peer review and to provide appropriate recognition for Early Career

Researchers who co-review manuscripts.

We thank the Reviewers for their constructive feedback on our manuscript entitled “RBM20 isoform regulation by independent transcription start sites adapts alternative splicing in development and disease.” We revised the manuscript accordingly and believe that the changes have substantially strengthened the work.

Summary of major changes in the revised manuscript:

- Added new WT and RBM20 lacZ-HOM Ribo-seq data with P-site mapping to support translation initiation at the AUG in exon 2 (new Fig. 2c,d; updated Results/Methods).
- Added experiments and analyses to clarify RBM20 proteoform localization and functional redundancy (fluorescently tagged isoforms; colocalization quantification; updated text and Supplementary figures).
- Expanded analysis of disease-specific regulation by integrating transcription factor programs, promoter motif scans, and promoter-variant burden (new Results section and new Fig. 6).
- Performed Gene Ontology enrichment on differentially spliced genes in HCM/DCM and added the results as Supplementary Fig. S5 d,e.
- Improved transparency and clarity throughout: expanded Methods (sequencing and bioinformatics), clarified figure legends (sample sizes/ages/abbreviations), and provided additional blot/gel context (including size markers and full images where applicable).

All changes are incorporated in the revised manuscript.

REVIEWER COMMENTS

Reviewer #1 (Remarks to the Author):

The authors of this manuscript identified an alternative transcription start site in canonical exon of RBM20, a well known splicing regulator. Mutation in RBM20 cause cardiomyopathies and understanding the regulation and function of Rbm20 of relevance. The manuscript is well written, the data sound and novel. Addressing a few comments would further increase the quality of the manuscript.

1) Validation of translation initiation at the ATG at exon 1B. The data presented in figure 2b are not completely convincing. The authors could include additional ribosomal sequencing experiments using harringtonine to better map the initiation on the ATGs and include ribosomal sequencing data from the lacZ-Hom mice to show translation initiation

Response:

We agree that direct evidence for the translation initiation site is important. To address this, we generated additional ribosome profiling (Ribo-seq) data from WT and RBM20 lacZ-HOM hearts and performed P-site mapping.

In both genotypes, we observe a prominent P-site peak at the AUG in exon 2, consistent with translation initiation at this codon. The lacZ-HOM model provides genetic support for this conclusion because it cannot produce an exon-1-initiated proteoform, yet still shows clear translation initiating at exon 2.

These data are now included in the revised manuscript (Fig. 2c, d) and described in the Results and Methods sections.

2) What is the function of the P-rich n-terminal region in the canonical isoform. The authors speculate that it may be more efficient at organizing RBM20 into nuclear speckles. Functional data supporting this hypothesis would strengthen the manuscript.

Response:

At present, isoform-discriminating antibodies are not available; our RBM20 antibody recognizes the shared C-terminus and therefore detects both proteoforms. To address this point experimentally, we generated fluorescently tagged constructs of the canonical and alternative RBM20 isoforms and assessed their subcellular localization in mammalian cells.

Both isoforms showed comparable nuclear localization and high spatial overlap when co-expressed. We have added quantification of this overlap and toned down speculative statements regarding isoform-specific nuclear compartmentalization.

To further address the reviewer's question, we added a cross-species sequence comparison (new Fig. 7a) showing that, compared with the shorter alternative B proteoform, the canonical proteoform retains a mammal-enriched proline-rich N-terminal stretch, whereas the sequence context surrounding the alternative AUG in exon 2 is highly conserved across vertebrates. We revised the Discussion to reflect that we do not detect consistent isoform-specific differences in subnuclear localization or splicing target choice; instead, the most robust difference is diversified, promoter-specific regulation of proteoform abundance, which provides an additional lever to increase RBM20 protein output during high-demand contexts (e.g., around birth and in HCM).

3) What drives upregulation of the alternative isoform in HCM vs upregulation of the canonical form in DCM? Could the authors use existing data on HF patient to look at regulatory elements that might explain alternative TSS usage in different forms of HF?

Response:

To explore potential drivers of disease-specific RBM20 isoform regulation, we expanded our analysis of the HCM and DCM cohorts. Specifically, we (i) analyzed differential expression of transcription factors, (ii) scanned the canonical and alternative RBM20 promoter regions for high-confidence transcription factor binding motifs, and (iii) performed RNA-seq-based variant calling in the promoter regions to assess cis-regulatory influences.

These results are now presented in a new Results section ("Disease-specific transcription factor programs and promoter variants shape RBM20 isoform expression") and summarized in a new Figure 6.

Reviewer #2 (Remarks to the Author):

In this manuscript, Radke et al. report the identification of a novel transcription start site in the RBM20 gene, leading to the production of a shorter protein isoform. Using RNA-seq technology and reanalysis of previously published datasets, the authors demonstrate that this isoform is expressed in both rat and human tissues, retains splicing activity, and exhibits differential regulation compared to the canonical isoform in the context of human disease. The manuscript is generally well written; the Methods section is thorough and appropriately detailed, and the figures are clear and support the authors' conclusions. While the discovery of this novel isoform provides valuable insight into an additional layer of RBM20 regulation, the study lacks experimental evidence directly addressing the biological significance or functional impact of the new isoform.

Minor Comments

1. There is a repeated paragraph between lines 71–73 and lines 78–81.

Response:

Thank you for your attention, we have removed the duplicate sentence (lines 78-81).

2. Figure 1b is not referenced in the main text.

Response:

We now reference Fig. 1b in the main text in the section describing generation and genotyping of the lacZ model, and we adjusted the figure legend accordingly.

3. The terminology used to refer to the different genetic models is somewhat inconsistent and occasionally confusing. It should be standardized throughout the manuscript to improve readability and interpretation

Response:

We agree and standardized nomenclature throughout the manuscript. We now consistently refer to the genotypes as lacZ-HET/lacZ-HOM and RRM-HET/RRM-HOM, and we updated the text and figure legends accordingly.

We used lacZ-HOM and lacZ-HET for RBM20 lacZ homozygote and heterozygote transgene mice, and RRM-HOM and RRM-HET for the RBM20 RRM knockout homozygote and heterozygote mice. We corrected the typos where we used different spellings.

Major Comments

1. In Figure 1f and Supplementary Figure 1a, the WT samples show two bands, with the smaller band appearing to match the size of the one detected in the lacZ mouse. Do the authors interpret this band as representing the alternative Rbm20 isoform? If so, what is the rationale for emphasizing the functional significance of this isoform if it is also present in WT animals?

Response:

We agree that the interpretation of the two RBM20 bands should be made explicit. In WT hearts, our C-terminal RBM20 antibody detects two bands, which is consistent with co-expression of a longer (exon 1-initiated) and a shorter (exon 1B-initiated) RBM20 proteoform. In lacZ-HOM hearts, only the shorter band is detected.

Importantly, the significance of the alternative proteoform is not that it is unique to the lacZ model, but that it is produced from an independent promoter/TSS and therefore can be regulated separately from the canonical transcript. This diversified transcriptional control provides a mechanism to adjust total RBM20 output (and thereby splicing capacity) in a context-dependent manner, as supported by the developmental datasets and by the preferential upregulation of the alternative isoform in human HCM.

We clarified this interpretation in the Results and figure legend, and we ensured that molecular weight markers and full blot images are provided for transparency (including the full blot for Fig. 1f in the Supplement).

2. In Figures 1h and 1l, it is unclear whether the nuclear localization patterns of the canonical and alternative isoforms are similar or distinct. The authors should clarify this point.

Response:

We agree that endogenous isoform-specific localization cannot be resolved with the available antibody because it recognizes the shared C-terminus. However, two independent approaches support similar nuclear localization of the short isoform: (i) immunofluorescence in lacZ-HOM tissue, where only the short isoform is present, and (ii) co-expression of fluorescently tagged canonical and alternative RBM20 constructs, which shows strong nuclear colocalization.

In addition, we fused now human RBM20 canonical to EGFP and human RBM20 altstart to mCherry and coexpressed both in mammalian HEK cells. Both Isoforms colocalize nearly complete (Pearsons' R 0,6;), so we expect an equal localization of both native isoforms of RBM20. We added this information to Supplementary Fig 2g. These data support a shared nuclear targeting and subnuclear compartmentalization of canonical and alternative RBM20; thus, the proteoform difference most plausibly alters effective dose and/or interaction topology rather than switching nuclear versus cytosolic localization.

3. In Figure 1i, the authors suggest that the two bands observed in the lacZ lane correspond to those in the RRM+/- mouse. However, the bands in the lacZ sample appear clearly lower in molecular weight. This discrepancy should be addressed.

Response:

We revised the description of the titin gel in Fig. 1i and clarified band identities. Importantly, lacZ-HOM hearts show an intermediate titin splicing phenotype compared with WT and RRM-HOM, consistent with partial RBM20 activity in the lacZ model.

To avoid overinterpretation based on band appearance, we updated the manuscript text and legend to emphasize the dose-dependent nature of the phenotype and to describe the control lanes used for size comparison.

4. In Supplementary figures 1c and 1d, the authors make similar claims regarding band identity and size, but appropriate control samples (e.g., RRM+/-) are missing for comparison.

Response:

We agree that additional controls strengthen the interpretation. We therefore included additional titin gels including RRM-HET samples for direct comparison, updated the corresponding Supplementary figures, and adjusted the text to reflect these controls.

5. Table 1 indicates improved ventricular output in the LacZ-HOM mouse. However, it is unclear whether this phenotype is due to reduced expression of the canonical RBM20 isoform or the gain of function from the alternative isoform. The authors should discuss this distinction more explicitly.

Response:

We apologize for the confusion. Table 1 does not indicate improved function in lacZ-HOM mice; rather, the data show prolonged relaxation and contraction times and a higher myocardial performance index, consistent with impaired global performance compared to WT.

Importantly, our interpretation is not that a unique function of the canonical isoform is lost (or that the alternative isoform gains a new function), but that the lacZ allele changes RBM20 proteoform composition and total effective RBM20 activity, resulting in an intermediate phenotype that is less severe than complete loss of RBM20 function in the RRM-KO.

We clarified this interpretation in the Results/Discussion and in the table description.

6. In all presented Ribo-seq datasets, the translational signal for the alternative Rbm20 isoform is higher than that of the canonical isoform. How do the authors reconcile this with their observation that the canonical isoform is more abundant than the alternative in most models except the LacZ-HOM?

Response:

This is an important point. Because translation of both proteoforms initiates at the shared AUG in exon 2, ribosome footprints over exon 2 and downstream coding regions cannot be unambiguously assigned to one proteoform after initiation.

Accordingly, we use Ribo-seq primarily to localize the dominant initiation site (AUG in exon 2) and to assess P-site patterns around this codon. We clarified this limitation in the Results and Methods and added new WT and lacZ-HOM Ribo-seq/P-site analyses (Fig. 2c,d).

7. If the values shown above the arcs in the Sashimi plot represent the number of splicing events observed, how do the authors explain that the value supporting the alternative isoform in WT samples is higher than that for the canonical isoform? This seems to contradict other expression data presented.

Response:

The numbers shown in sashimi plots reflect junction-spanning reads and are influenced by sequencing depth and normalization. To quantify isoform usage more robustly, we report normalized junction counts and PSI values in addition to the sashimi visualization.

We clarified this in the manuscript and expanded the description of how junction counts and PSI values were computed.

8. Although the authors present a PSI plot, it remains unclear whether they are claiming that the alternative RBM20 isoform regulates a distinct Ttn splicing pattern compared to the canonical isoform. This should be clarified with supporting data or discussion.

Response:

We agree that this should be stated more clearly. We do not claim that the alternative RBM20 proteoform drives a distinct titin splicing program. Instead, our data support a model in which the alternative proteoform retains splicing activity and contributes to overall RBM20 dosage/functional capacity.

We revised the Discussion accordingly (and removed/tempered statements that could be read as isoform-specific splicing regulation).

9. Performing Gene Ontology analysis on the differentially spliced genes shown in Figures 5i and 5l would provide further insight into the molecular pathways through which the alternative RBM20 isoform may contribute to cardiomyopathy.

Response:

We performed Gene Ontology enrichment on the differentially spliced gene sets in HCM and DCM and added the results as Supplementary Fig. S5d and e. In HCM, enriched terms cluster around actin cytoskeleton and adhesion/junction biology (e.g., actin binding, focal adhesion, cell–cell junction), whereas DCM shows enrichment for muscle differentiation/contractile programs together with RNA splicing and transcription co-regulator activity. We now summarize this interpretation in the Results

section ('Regulation of RBM20 isoforms in cardiac disease', after the HCM and DCM splicing analyses) and briefly integrate it into the Discussion to clarify the biological meaning of the GO outputs.

10. While RBM20 is predominantly expressed in cardiomyocytes, it would be valuable to validate the findings shown in Figure 5 using publicly available human single-cell RNA-seq datasets for hypertrophic and dilated cardiomyopathy. This could help exclude the possibility that the observed effects are driven by non-cardiomyocyte populations.

Response:

We agree that bulk RNA-seq can reflect both transcriptional regulation and cell-composition effects. We therefore clarified in the manuscript that our isoform analyses use splice-junction evidence (exon-1/exon-1B to exon-2 junction counts), which remains informative even under modest compositional shifts. To address the reviewer's request for cellular context, we cite and discuss published single-nucleus and single-cell human heart atlases that show RBM20 enrichment in cardiomyocytes and broad conservation across ventricular cardiomyocyte populations, and we explicitly acknowledge that disease-associated changes in cell composition can contribute to bulk RBM20 abundance changes. We checked RBM20 expression across cardiac cell types using publicly available single-cell expression resources (Human Protein Atlas), which show RBM20 to be strongly enriched in cardiomyocytes compared with non-myocyte populations - in Myocytes (>2100nCPN), while the next cell line with RBM20 expression are neuronal cells with a magnitude lower expression (<160nCPN) other cell types show lower or no RBM20 expression.

We added a short statement to the manuscript to support this point and we also note in the Discussion that bulk-tissue analyses cannot fully exclude contributions from shifts in cell composition.

11. The manuscript presents convincing data regarding the expression of the alternative Rbm20 isoform. However, its functional role in cardiac development and disease progression remains insufficiently explored. This limits the overall impact of the study. Additional functional experiments or a more thorough discussion of its biological relevance would be beneficial.

Response:

We agree that the biological interpretation should be strengthened. To address this, we expanded the Discussion to focus on (i) how independent promoter/TSS usage provides an additional regulatory axis to tune RBM20 activity during the perinatal window and in disease, and (ii) why a transcriptional switch between promoters can change RBM20 output without requiring changes in the coding sequence.

Experimentally, we added/expanded analyses that support functional redundancy of the proteoforms at the level of subcellular localization and splicing readouts: fluorescently tagged isoforms show strong nuclear colocalization, and key RBM20 targets (e.g., titin, Camk2d) remain largely spliced in the lacZ model with reduced efficiency.

Finally, we emphasize that the main robust phenotype we observe in vivo is an intermediate RBM20-activity state (lacZ) compared to complete functional loss (RRM-KO), consistent with partial compensation by the alternative B proteoform.

We also clarified what our experiments do and do not support. The alternative B proteoform (i) localizes to the nucleus, (ii) overlaps strongly with canonical RBM20 when co-expressed, and (iii) supports splicing of established RBM20 targets in vivo, as evidenced by preserved (though less efficient) splicing patterns in the lacZ model. In contrast, we currently do not observe a robust isoform-specific re-localization or a qualitatively distinct splicing target set in our transcriptome-wide analyses.

Finally, we emphasize that the most robust in vivo signature is an intermediate RBM20-activity state in lacZ mice (relative to complete functional loss in the RRM-KO), consistent with partial compensation by an independently regulated, shorter proteoform that increases RBM20 dosage but does not fully recapitulate canonical output.

Reviewer #3 (Remarks to the Author):

The manuscript by Radke et al. with the title „RBM20 isoform regulation by independent transcription start sites adapts alternative splicing in development and disease“ identified an unrecognized transcription start site (TSS) for RBM20 via genetic perturbations in mice. The study convincingly shows that the ratios of the different RBM20 isoforms are controlled during development and are contrary altered in DCM versus HCM. The authors furthermore focus on newly and already published sequencing data from mice, rat and humans, and provide novel insights into the landscape of RBM20 isoform regulation.

The manuscript contains the information that is required to understand the conducted experiment, the used approach is well described and allows others to perform additional mechanistic studies also in other disease-relevant contexts. To fully agree to a publication in Nature Communications, I suggest the following optimizations:

Major:

1. The authors used a self-produced RBM20 antibody, which is not described nor referenced throughout the manuscript. Therefore, not enough detail is provided for the work to be reproduced. Ideal would have been a comparison to already commercially available RBM20 antibodies.

Response:

We apologize for the missing information. The self-produced RBM20 antibody used in this study was generated and validated previously (Guo et al., Nat. Med. 2012). We added the corresponding reference and details (antibody source/epitope information) to the Methods and the Supplementary antibody tables.

In addition, we evaluated commercial RBM20 antibodies; however, under our conditions they did not provide a specific signal comparable to our in-house antibody.

For additional information, the commercial antibody PA5-57404 was replaced by PA5-58068 which was not working in our hands in IF (mouse). Also R35345 did not work in our hands in IF (mouse); HPA035806 only recognize human but not mouse

2. The introduction as well as the abstract state strongly that RBM20 is regulated by phosphorylation. However, it is then neglected through the manuscript. The authors see a size shift of the shorter RBM20 1B isoform, but don't state if the observed size shift is expected by an exclusion of the first exon. The deletion should be around 128 amino acids (~13 kDa), however it seems that the shift on the provided Western blot is bigger than that. A quick search at the phosphosite plus database indicates two potential phosphoserines in the first part of RBM20 for humans, could that explain the bigger than 13 kDa shift? The authors also neglect that the first exon includes the L-rich domain of RBM20, the manuscript only states that the P-rich domain is deleted in 1B.

Response:

We agree that phosphorylation should be presented with appropriate context as our work does not investigate the contribution of posttranslational modification, but focusses on alternative isoform expression. We revised the Abstract/Introduction to clarify that phosphorylation has been reported to regulate RBM20 subcellular localization and splicing activity, whereas the isoform selection described in our study is driven by alternative transcription initiation.

We also clarified expectations regarding the size difference between the two proteoforms (based on sequence) and noted that the apparent migration on SDS-PAGE can indeed deviate from predicted molecular weight, potentially due to low-complexity regions and/or post-translational modifications as suggested by the reviewer.

We also address the reviewer's P-rich domain question explicitly in the Discussion and by clarifying domain architecture in the Results. RBM20 contains a proline-rich low-complexity segment encoded by exon 1 and a leucine-rich segment in exon 2; therefore, the exon 1B-driven proteoform lacks the exon 1 P-rich N-terminus and initiates within the exon 2 L-rich sequence (Fig. 7a). Current reviews of RBM20 emphasize that no definitive molecular function has yet been assigned to these compositionally biased N-terminal regions, but low-complexity segments in splicing factors frequently modulate interaction valency, condensate behavior, and proteoform turnover. In our study, both canonical and alternative RBM20 localize to nuclear puncta and co-localize upon co-expression (Supplementary Fig. S2g), whereas the alternative isoform shows markedly reduced protein output relative to its mRNA (Supplementary Fig. S2h-j). Together, these data support the conservative conclusion that the missing P-rich N-terminus primarily tunes effective RBM20 dose (via proteoform output and/or stability) rather than driving a qualitatively different subnuclear localization under basal conditions.

3. The authors show convincingly that the regulation of isoforms is not mediated via translational regulation, and assume that it comes from a transcriptional regulation via alternative transcription start sites. The authors then suggest quickly that the ratio maybe regulated by two independent promoters for both transcript start sites. This concept needs some further details to be fully incorporated into the main text. However, could it also be that the integration of the LacZ gene into Exon1 disrupts normal splicing of RBM20 itself and therefore a new splice isoform with only Exon 2 is incorporated via a non-canonical new splice site? This should be quickly discussed as an alternative possibility or experimentally excluded in the main text.

Response:

We agree that this alternative explanation should be excluded. Multiple lines of evidence argue that the alternative transcript is not an artifact of the lacZ knock-in: we detect exon 1B junction reads in WT mice and in independent datasets, and we also observe the alternative isoform in rat and human RNA-seq data.

We clarified this logic in the Results/Discussion and explicitly state that lacZ-HOM mice serve as a genetic tool to reveal the alternative proteoform, but the underlying alternative promoter/TSS is present in WT.

Our ribo Seq data demonstrates that RBM20 has at least 2 independent promoters and produces a canonical isoform starting in exon 1 and a new identified isoform starting in exon 2. The translation start site for the new alternative isoform is starting in exon 1B which was explored in Race PCR. This new identified alternative isoform of RBM20 is also expressed in the wildtype (see experimental data in Figure 2, 4 and 5)

4. In general, the manuscript does not describe prominent observed bands on provided Western blots and agarose gels, which should be discussed or contextualized. The authors state that Filippello et al. already described an independent translation initiation site in Exon 2 of Rbm20, but also state that Filippello did not detect the here called full-length canonical isoform of RBM20 in human cardiac tissues. The here provided RACE PCR results in supplementary Fig. 2e show that the shorter fragment seems to be the predominant form as well, which disagrees to the other shown data in the main figures. This needs further clarification. In addition to exons 1 and 1B, the authors also identified an alternative TSS upstream of exon 1, termed exon 1A, however the shown RACE PCR gel results do not detect a longer variant, which should be discussed in the main text too. In general, all shown western blots and agarose gels should indicate the ladder/size at the side! The Western blot in Fig. 1f shows only the bigger band (please provide the full picture, and show also the shorter bands), the sashimi plot in Fig. 2c indicates the shorter version should be predominant. The Western blot in supplementary Fig. 2a also shows mostly that the bigger band is predominant. This needs some better explanations in the main text

Response:

We agree that gel- and blot-based results require clearer description and context. We expanded the Results text and the figure legends to explain RBM20 bands and PCR products, added molecular

weight/size markers, and provided uncropped original blots for key panels in the Supplement. Importantly, 5' RACE identifies transcription start sites but provides only semi-quantitative band intensities due to nested PCR amplification bias, and we therefore quantify isoform usage using splice-junction counts/PSI values and RT-qPCR (Fig. 1k; Fig. 2e). Finally, we added an explicit explanation that transcript abundance and proteoform abundance can decouple; our cell-based expression experiments support post-transcriptional regulation of alternative proteoform output (Supplementary Fig. S2h–j), which resolves the apparent discrepancy between exon 1B junction prominence and a weaker short RBM20 band on Western blots (Fig. 1f; Supplementary Fig. 2a). We also note that exon 1A remains below detection by gel-based 5' RACE, consistent with low abundance, while RNA-seq junction reads provide independent evidence for exon 1A-to-exon 2 connectivity (Fig. 2e).

5. Supplementary Fig. 2a shows different RBM20 bands in the provided Western blot, which indicates different isoforms distributions in the isolated muscle tissues, which needs to be further discussed in the main text. The LacZ mice seem to differ more between tibialis anterior muscle and quadriceps than the shown WT mice.

Response:

We agree that tissue-specific differences merit discussion. We added text to the Results/Discussion noting that the relative abundance of RBM20 bands varies across skeletal muscles, consistent with tissue-specific regulation of RBM20 expression and/or isoform usage.

Because skeletal muscle analyses were not the primary focus and sample sizes are limited, we avoid overinterpretation and present this as an observation motivating future work.

6. Fig.1 I needs quantification of the colocalization to make the claim that there is colocalization of RBM20 and U2AF65!

Response:

We added quantification of colocalization (Pearson's correlation coefficient) and revised the text to state that RBM20 partially colocalizes with U2AF65.

7. The introduction mentions that the new identified isoform of RBM20 might offer potential avenues for therapeutic intervention, without further details on possible intervention points in the main text.

Response:

We agree and expanded the Discussion to provide more concrete therapeutic context. RBM20 is an established target to modulate titin compliance, for example via partial RBM20 inhibition with antisense oligonucleotides. Recent work optimized RBM20-ASO dosing in a cardiometabolic HFpEF mouse model

and demonstrated improved diastolic function through selective shifts toward more compliant titin isoforms (Methawasin et al., *Cardiovasc Res* 2025; PMID: 41104480). In parallel, genome editing approaches are emerging to correct pathogenic RBM20 variants, including prime editing to correct the RBM20 P633L mutation in human iPSC-derived cardiomyocytes with molecular phenotypic rescue (Roman et al., *Mol Ther Nucleic Acids* 2025; PMID: 41210585).

Our identification of independent promoter/TSS usage suggests an additional potential intervention point: in principle, modulating promoter choice could shift proteoform output and total RBM20 activity without directly altering the RBM20 coding sequence. This concept is supported by broader evidence that alternative transcription start site usage can tune RNA processing outputs and is frequently re-wired in disease (Alfonso-Gonzalez & Hilgers, *Trends Cell Biol* 2024; PMID: 38531762).

We added this perspective to the Discussion and updated the relevant text in the Introduction.

Minor:

1. No reference in the main text to Fig 1b

Response:

We have added a reference to Fig. 1b in the main text (see also our response to Reviewer #2, point 2).

2. All figure legends should contain and explain all used abbreviation, e.g. what is “rec” in Fig. 1b

Response:

We added the explanation for the abbreviation in the figure legends. Rec stands for homologous recombination after FLP-Recombinase expression and is used to determine the genotyped allele.

3. Fig. 1K shows that exon 2-3 is as highly expressed in LacZ as in WT! Can the authors comment on the here observed compensation mechanism?

Response:

We agree that the apparent preservation of exon 2–3 signal in lacZ compared with WT is consistent with partial compensation at the transcript level. This is also supported by our junction-count and PSI analyses, which indicate increased usage of the alternative start exon(s) in lacZ.

We also clarified an important technical point: RT-qPCR values are comparable within a primer/probe set (WT vs lacZ) but should not be interpreted as absolute comparisons across different assays targeting different exon junctions. We revised the text/legend accordingly.

4. All figure legends should indicate the number and age of the used mice or the corresponding used data sets, e.g. Fig. 2b does not indicate this, but it is required to understand at which time point the alternative Rbm20 transcription starting site (TSS) in intron 1 is detected by mRNA-seq and Ribo-seq. Same for Fig. 1h, the Immunofluorescence localization of RBM20 in nuclei of WT and lacZ hearts does not indicate the age.

Response:

We have added/expanded information on sample age, group sizes, and the source of external datasets in figure legends and/or the Methods. For example, immunofluorescence in Fig. 1h is from adult mice at 100 days of age, and panels based on published RNA-seq/Ribo-seq datasets are now explicitly attributed in the Methods and/or figure legends.

5. What is the difference between “RiboSeq” and “Psite” data in Fig. 2?

Response:

Ribo-seq coverage tracks show the distribution of ribosome-protected fragments across transcripts, whereas P-site tracks represent inferred ribosomal P-site positions (the estimated codon occupying the ribosomal active site) and therefore provide a higher-resolution view of translation along the coding sequence. We clarified this in the figure legend and Methods.

6. In Supplementary Fig. 2f, what does the axis label “Norm to H” mean?

Response:

“Norm to H” indicates that expression values were normalized to heart. We added this explanation to the Supplementary Fig. 2f legend.

7. The **material and methods part is quite sparse** in regard of the performed mRNA seq procedure and could benefit from a more detailed description of used materials/kits.

Response:

We expanded the Materials and Methods to provide a more detailed description of library preparation, sequencing, and bioinformatic processing, including tools, versions, and key parameters.

8. Figure legend 4 e should be labeled with “f” and not “e”

Response:

We checked the panel labels and clarified the Figure 4 legend to explicitly describe what is shown in panels e and f. If any ambiguity remained, we adjusted the legend so that labels and descriptions match unambiguously: **e)** *Rbm20* expression level around birth in the rat **f)** *Rbm20* expression levels around birth based on reads mapping to the canonical (blue), alternative B (red) in rat”

9. The method section states that the study was performed only on male mice, it would have been interesting to see if there is a sex difference in the RBM20 isoform ratio. Why has this not been considered?

Response:

We used male mice to minimize variability and to keep animal numbers as low as possible for the extensive phenotyping and sequencing experiments performed. We agree that sex-dependent differences in RBM20 regulation is an important question. We now state this explicitly as a limitation and future direction in the Discussion and clarify in Methods that we used sex-matched cohorts but did not perform sex-stratified analyses because the study was not powered for genotype-by-sex effects.

Reviewer #4 (Remarks to the Author):

Response:

Thank you for taking the time and effort to review our manuscript.